

# Strangeness neutrality and QCD thermodynamics

**Wei-jie Fu[1], Jan M. Pawlowski[2] and Fabian Rennecke[3⋆]**

**1** School of Physics , Dalian University of Technology, Dalian, 116024, P.R. China
**2** Institut für Theoretische Physik, Universität Heidelberg,
Philosophenweg 16, 69120 Heidelberg, Germany
**3** Physics Department, Brookhaven National Laboratory, Upton, NY 11973, USA

⋆ frennecke@bnl.gov

## Abstract

Since the incident nuclei in heavy-ion collisions do not carry strangeness, the global net strangeness of the detected hadrons has to vanish. We investigate the impact of strangeness neutrality on the phase structure and thermodynamics of QCD at finite baryon and strangeness chemical potential. To this end, we study the low-energy sector of QCD within a Polyakov loop enhanced quark-meson effective theory with 2+1 dynamical quark flavors. Non-perturbative quantum, thermal, and density fluctuations are taken into account with the functional renormalization group. We show that the impact of strangeness neutrality on thermodynamic quantities such as the equation of state is sizable.

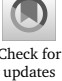
# 1   Introduction

Ultrarelativistic heavy-ion collisions performed at RHIC and LHC aim to explore the phase structure of quantum chromodynamics (QCD) at finite temperature and density. One of the key challenges is to extract properties of the quark-gluon plasma (QGP) created in such collisions from the hadronic final states that reach the detector. The success of hadron resonance gas models (HRG), which are based on thermal distributions of noninteracting hadrons, in describing various aspects of the hadronization process might suggest that the system at the time of freeze-out can be described by equilibrium thermodynamics characterized by temperature and chemical potentials [1].

Since the timescale of the weak interactions is much longer than the equilibration time of the strongly interacting QGP, quark number conservation of the strong interactions should hold from the initial stage up to the freeze-out. So the strangeness and charge/isospin of the incident nuclei determine the strangeness- and isospin chemical potentials $\mu_S$ and $\mu_I$ at freeze-out. For instance, the absence of strange quarks in nuclei implies strangeness neutrality, i.e. the net strangeness has to vanish. The baryon chemical potential $\mu_B$, which is directly related to the baryon number at central rapidity, additionally depends on the energy of the collision because the rapidity distributions of net-baryon number show a distinctive beam-energy dependence. In fact, this is the basis of current and future beam-energy scan experiments aimed at exploring different region of the QCD phase diagram [2–7].

To understand the properties of matter created in heavy-ion collisions it is therefore indispensable to take these constraints into account. Since quarks, mesons and baryons can carry finite strangeness and isospin, the details of how these constraints are fulfilled depend crucially on the state of QCD matter. Understanding this from a theoretical point of view poses many challenges. The different phases of QCD, including the dynamics of quarks, gluons and hadrons at various temperatures and chemical potentials need to be captured. Hence, purely hadronic effective models which are only valid at the lowest energies and QCD perturbation theory only valid at very high energies are only of limited use. Owing to the notorious sign problem at finite $\mu_B$, lattice QCD simulations are restricted to vanishing chemical potential. Nonetheless, tremendous progress has been made in recent years in exploring the QCD equation of state at finite $\mu_B$ on the lattice through, e.g. the Taylor expansion of the thermodynamical potential as a function of $\mu_B/T$ around $\mu_B = 0$ [8] or the analytic continuation from imaginary chemical potential [9], among many other approaches [10, 11]. These techniques allowed first studies of the freeze-out conditions of heavy-ion collisions subject to the constraints on strangeness and isospin on the lattice [12–14]. Since both methods rely on expansions in powers of $\mu_B/T$, exploring regions of the phase diagram with $\mu_B/T \gtrsim 1$ on the lattice might entail unknown and potentially large systematic errors. For instance, at small beam energies at RHIC the HRG

predicts $\mu_B/T > 2$ at the freeze-out [15], so current experiments probe regions of the phase diagram where state-of-the-art first principle methods might not be fully reliable.

Functional continuum methods, such as the functional renormalization group (FRG) and Dyson-Schwinger equations (DSE) do not suffer from the sign problem, so the inclusion of finite chemical potential is possible without the corresponding systematic errors. A lot of progress has been made towards the study of QCD from first principles, e.g. [16–23] and references therein. However, due to the necessity of truncating the effective action of QCD, results at finite chemical potential from first principles are currently only accessible with unknown and potentially large systematic errors. Functional continuum methods are in some sense complementary to the lattice, since the most common sources of systematic errors on the lattice, such as finite-size effects, chiral fermions and the sign problem, are not present in continuum methods and, vice versa, the lattice does not have to rely on truncations of the effective action.

Low-energy effective theories of QCD have proven time and again that they can provide valuable insights on the QCD phase structure. Their strength lies in the potential to identify physically relevant effects that prevail also in the full theory. Prominent examples relevant for the present work are Polyakov loop enhanced Nambu–Jona-Lasinio models (PNJL), Polyakov loop enhanced quark-meson models (PQM) and (the closely related) chiral matrix models. They can be constructed to share the same global symmetries as QCD and exhibit similar or even the same symmetry breaking patterns as the chiral transition of QCD. Owing to the coupling to a non-vanishing gluon background field, the deconfinement transition can also be captured in a statistical manner [24]. In mean-field approximations, the phase structure and thermodynamics of QCD have been studied in great detail with these models, see e.g. [25–33] and [34] for a recent review. In this context, the constraint of strangeness neutrality has first been imposed in the study of the phase structure in [35]. In compliance with expectations from the HRG [1] and the lattice [12], it was demonstrated that a finite strangeness chemical potential is necessary to ensure strangeness neutrality at finite temperature and baryon chemical potential. This is related to the intricate interplay of quark, meson and baryon effects mentioned above.

Concerning strangeness and isospin dynamics, a major shortcoming of mean-field studies is the lack of dynamics of the most relevant degrees of freedom in the hadronic phase. Owing to their nature as pseudo Goldstone bosons of spontaneous chiral symmetry breaking, these are certainly pions and kaons regarding the effects related to isospin and strangeness. It is therefore conceivable that their quantum fluctuations have to be accounted for in order to accurately describe the QCD medium as generated in heavy-ion collisions. A major challenge is that hadronic fluctuations are in general of non-perturbative nature. The FRG has been proven to be very useful here, since it allows for the non-perturbative regularization and renormalization of quantum fluctuations in low-energy models. For two flavors, the phase structure and thermodynamics of (P)QM models have been studied exhaustively with the FRG, e.g. [36–46]. These works carved out the crucial role of meson fluctuations in the QCD equation of state. Finite isospin chemical potential has been investigated in [47] within a QM model. However, the constraint on isospin from heavy-ion collision has not been considered in this work. Strangeness requires at least three flavors. In this case, first studies of the phase structure with the QM beyond mean-field have been carried out in [48–50] and the PQM at vanishing density has been studied in [51]. Fluctuations in the strange sector have been shown to be quantitatively and qualitatively relevant for the phase structure of QCD in the former works. In the latter work it has been demonstrated that lattice thermodynamics at vanishing density can be reproduced by including fluctuations into the PQM model with the FRG.

In this work we extend the previous works in two ways. The first is the extension of [51] to finite baryon chemical potential $\mu_B$ and the confrontation of the results on the equation of state with most recent lattice results at finite $\mu_B$. Second, and most importantly, we introduce a finite strange chemical potential $\mu_S$ and derive the corresponding functional renormalization group

equations for the 2+1 flavor PQM. This allows us to impose the strangeness neutrality condition on the equation of state in terms of a $T$- and $\mu_B$-dependent $\mu_S$. As discussed above, this is a property imprinted on the matter created in heavy-ion collision from its initial conditions. For the first time, we are able to study the influence of strangeness neutrality on the thermodynamics and phase structure of QCD beyond mean-field directly at finite baryon chemical chemical potential. Genuine finite density effects related to the dynamics of strange hadrons are accessible this way. This is of relevance for a general understanding of the properties of strongly interacting matter as created in heavy-ion collisions.

This paper is organized as follows: In Sec. 2 we introduce the effective low-energy model used here, including a discussions of the coupling of mesons to $\mu_S$ and the finite gluon background. The functional renormalization group and the derivation of the corresponding renormalization group equations is discussed in Sec. 3. We present our results in Sec. 4. After the discussion of the initial conditions for the solution of the RG equations in Sec. 4.1, we check the validity of our model by comparing it to lattice results at vanishing and finite $\mu_B/T$ in Sec. 4.2. In Sec. 4.3 we determine the strangeness chemical potential neccesary to fulfill the strangeness neutrality condition and discuss the role of quark, meson and baryon dynamics for our results. In Sects. 4.4 and 4.5 we discuss the influence of strangeness neutrality of the thermodynamics and the phase structure of QCD. A summary and a brief outlook are given in Sec. 5. Details on the parametrization of the Polyakov loop potential, thermodynamics at large $\mu_B$ and the initial conditions are provided in the appendices.

## 2 $N_f = 2 + 1$ QCD at low energies

Here we discuss the construction of a low-energy effective theory of QCD that allows us to describe the main features of strangeness and the phase structure on the same footing.

### 2.1 Chemical potentials

In QCD the numbers of each flavor are conserved separately. So in general there is an independent chemical potential for each quark flavor, e.g. [52],

$$\mu_u \bar{u}\gamma_0 u + \mu_d \bar{d}\gamma_0 d + \mu_s \bar{s}\gamma_0 s \,. \tag{1}$$

The quark chemical potentials can be rewritten in terms of baryon-, strangeness- and isospin chemical potentials as follows

$$\mu = \begin{pmatrix} \mu_u \\ \mu_d \\ \mu_s \end{pmatrix} = \begin{pmatrix} \frac{1}{3}\mu_B + \frac{1}{2}\mu_I \\ \frac{1}{3}\mu_B - \frac{1}{2}\mu_I \\ \frac{1}{3}\mu_B - \mu_S \end{pmatrix} \,. \tag{2}$$

We remark that on the lattice the quark chemical potentials are typically written in terms of baryon-, strangeness- and charge chemical potentials, leading to

$$\mu = \begin{pmatrix} \frac{1}{3}\mu_{B,\text{lat}} + \frac{2}{3}\mu_Q \\ \frac{1}{3}\mu_{B,\text{lat}} - \frac{1}{3}\mu_Q \\ \frac{1}{3}\mu_{B,\text{lat}} - \frac{1}{3}\mu_Q - \mu_{S,\text{lat}} \end{pmatrix} \,, \tag{3}$$

see e.g. [12–14]. Comparing the two definitions we infer that $\mu_I = \mu_Q$ while $\mu_{B,\text{lat}} = \mu_B - 1/2\mu_Q$ and $\mu_{S,\text{lat}} = \mu_S - 1/2\mu_Q$. Note however, that $\mu_B, \mu_{B,\text{lat}}$ couple to the same operator $\bar{q}\gamma_0 q$ and

baryon number fluctuations are either described with derivatives w.r.t. $\mu_B$ or $\mu_{B,\text{lat}}$. Moreover, for $\mu_I = \mu_Q = 0$ the two definitions agree.

Hadrons carry charges associated to these chemical potentials, and hence couple to the quark chemical potential $\mu_q$. This coupling naturally emerges in the functional renormalization group approach from an evolution of QCD from large momentum to low momentum scales and the introduction of hadrons as effective low energy degrees of freedom via dynamical hadronization [53–55], see [22,23,56–58] for applications to QCD. The coupling of the chemical potentials to hadrons then follows directly from the Silver Blaze property of QCD [59]. At vanishing temperature, the chemical potential dependence of an Euclidean $n$-point function of fields $\phi_i$ with associate particle numbers $c_i$ is given by a simple shift of the external frequency [60,61]

$$p_{i,0} \rightarrow p_{i,0} + i c_i \mu_i. \tag{4}$$

Hence, one just needs to shift the frequencies of the kinetic terms in the effective action according to the Silver Blaze property.

In the present low energy effective field theory setup it is simpler to utilise a flavor symmetry argument, see e.g. [52]. At its core this argument carries the Silver blaze property of QCD discussed above, and it is straightforward to check that both constructions yield the same result. Concentrating on the mesons for the moment, we introduce the chemical potential as a vector source. Then the chemical potential in (1) is written as

$$C_\nu \equiv \delta_{\nu 0} C,$$

$$C \equiv \text{diag}\left(\frac{1}{3}\mu_B + \frac{1}{2}\mu_I, \frac{1}{3}\mu_B - \frac{1}{2}\mu_I, \frac{1}{3}\mu_B - \mu_S\right). \tag{5}$$

Using this in the full quark part of the QCD Lagrangian we arrive at

$$\mathcal{L}_q = \bar{q}\left(\gamma_\nu D_\nu + \gamma_\nu C_\nu\right)q = \bar{q}\gamma_\nu \bar{D}_\nu q, \tag{6}$$

with the modified covariant derivative $\bar{D}_\nu = D_\nu + C_\nu$ and $D_\mu = \partial_\mu - i g A_\mu$. This action is invariant under an extended local $U(N_f)$ flavor symmetry if the vector source $C_\nu$ transforms under local $U(N_f)$ transformations $\mathcal{U}(x)$ as

$$C_\nu \rightarrow \mathcal{U}(x) C_\nu \mathcal{U}^\dagger(x) + \mathcal{U}(x) \partial_\nu \mathcal{U}^\dagger(x), \tag{7}$$

not to be confused with chiral flavor rotations. Since the gauge part of the modified covariant derivative is flavor-blind, gauge invariance is trivially guaranteed. Scalar and pseudoscalar mesons are represented as entries of a flavor matrix in the adjoint representation of the flavor rotations defined in (7),

$$\Sigma = T^a(\sigma_a + i\pi_a). \tag{8}$$

Here the generators are $T^0 = \mathbb{1}/\sqrt{2N_f}$ and $T^{a \in \{1,\dots,N_f^2-1\}} \in SU(N_f)$. The meson sector inherits the local flavor symmetry of the quark sector as described above. Since the mesons transform in the adjoint representation, one can immediately write down the corresponding covariant derivative,

$$\bar{D}_\nu \Sigma = \partial_\nu \Sigma + [C_\nu, \Sigma]. \tag{9}$$

The chemical potential can be rewritten conveniently as

$$\mu = \frac{1}{3}\mu_B \mathbb{1} + \text{diag}\left(\frac{1}{2}\mu_I, -\frac{1}{2}\mu_I, -\mu_S\right). \tag{10}$$

With (10) and (9) it follows trivially that the baryon chemical potential does not couple to the mesons, as it should. In turn, mesons are sensitive to strangeness and isospin. In this work we assume light isospin symmetry and therefore set $\mu_I = 0$.

## 2.2 Low energy effective theory

Here we discuss the low energy effective theory in terms of its effective action. It has to captures the basic dynamics related to strangeness at low energies. Dynamically most relevant are the kaons, since they are pseudo Goldstone bosons with strangeness ±1. Chiral symmetry requires that if kaons are included in the effective action, all other mesons in the lowest scalar and pseudoscalar meson nonet have to be taken into account as well. This can be understood intuitively by considering the quark-antiquark scattering channels where the pseudoscalar kaons emerge as resonances,

$$\mathcal{L}_K \sim \left(\bar{u}\gamma_5 s\right)^2 + \left(\bar{d}\gamma_5 s\right)^2 + \left(\bar{s}\gamma_5 u\right)^2 + \left(\bar{s}\gamma_5 d\right)^2$$

$$\sim \left[\bar{q}\gamma_5(T^4 - iT^5)q\right]^2 + \left[\bar{q}\gamma_5(T^6 + iT^7)q\right]^2$$

$$+ \left[\bar{q}\gamma_5(T^4 + iT^5)q\right]^2 + \left[\bar{q}\gamma_5(T^6 - iT^7)q\right]^2, \tag{11}$$

where we choose the Gell-Mann matrices as $SU(N_f)$ generators. In terms of QCD flows for the effective action the four-fermi interactions including their momentum-dependent couplings emerge from gluon exchange diagrams. Note that the individual terms in Eq. (11) can in principle also have different couplings. However, it is straightforward to show that this channel explicitly breaks $U(N_f)_L \times U(N_f)_R$ chiral symmetry in any case. Since we are also interested in the phase transition, the only allowed sources of explicit chiral symmetry breaking are the current quark masses, otherwise chiral symmetry restoration cannot be captured properly. The four quark interaction channel that gives rise to a kaon resonance and respects chiral symmetry is

$$\mathcal{L}_K \subset \mathcal{L}_{4q} = \left(\bar{q}\, T^a q\right)^2 + \left(\bar{q}\, i\gamma_5 T^a q\right)^2. \tag{12}$$

Bosonizing this channel via a standard Hubbard-Stratonovich transformation [62,63], or selfconsistently with dynamical hadronization, yields an effective action containing the lowest scalar and pseudoscalar meson nonet as defined in Eq. (8), including their coupling to quarks. Note that Eq. (12) also contains the parity partners of the kaons, the kappas (or $K_0^*$), as additional open-strange mesons. Chiral symmetry dictates that we have to take them into account even though their mass is above 1 GeV so they are dynamically irrelevant. Resonances with the quantum numbers of pions, $\eta$, $\eta'$, $f_0(980-1370)$ and the critical modes of the chiral transition, the $\sigma$-mesons ($f_0(500)$), are also included in Eq. (12). Note however, that the identification of the heavy scalar meson is not entirely clear in our case since we find a mass of about 1150 MeV, which is between the known $f_0(980)$ and $f_0(1370)$ states. For more details on this construction see e.g. [49]. Including these dynamical mesons, their effective potential and coupling to quarks allows us to describe the chiral phase transition.

Statistical confinement is included via a (temporal) gluon background field $\bar{A}_\mu \equiv \bar{A}_0 \delta_{\mu 0}$ and a corresponding effective potential $U_{\text{glue}}(\bar{A})$. This is discussed in more detail in the next section. Putting all this together gives rise to a Polyakov loop enhanced quark-meson (PQM) model with 2+1 dynamical quark flavors at finite baryon and strangeness chemical potential. It is an approximation for the full effective action of low energy QCD valid below momentum scales $k \lesssim \Lambda$ with the ultraviolet cutoff scale $\Lambda \sim 1$ GeV. By definition $\Lambda$ is the scale below which gluons decouple from the matter sector of QCD, and hence constituent quarks and hadrons in a gluon background field provide a good description of QCD. We will elaborate on this further in Sec. 3.

In the current work we use the following approximation to the full scale-dependent Eu-

clidean effective action of the 2+1 flavor PQM model,

$$\Gamma_k = \int_x \left\{ \bar{q}\big(\gamma_\nu D_\nu + \gamma_\nu C_\nu\big)q + h\,\bar{q}\,\Sigma_5 q + \mathrm{tr}\big(\bar{D}_\nu \Sigma \cdot \bar{D}_\nu \Sigma^\dagger\big) + \widetilde{U}_k(\Sigma, \bar{A}) + U_{\mathrm{glue}}(\bar{A}) \right\}. \tag{13}$$

In (13) quantum, thermal and density fluctuations of modes with Euclidean momenta $\Lambda \geq |p| \geq k$ have been integrated out. The gauge covariant derivative is $D_\nu = \partial_\nu - ig\bar{A}_\nu$ and $\Sigma_5 = T^a(\sigma_a + i\gamma_5\pi_a)$. The effective meson potential $\widetilde{U}_k(\Sigma, \bar{A})$ consist of a fully $U(N_f)_L \times U(N_f)_R$ symmetric part plus pieces that explicitly break subgroups of the full chiral symmetry group,

$$\widetilde{U}_k(\Sigma, \bar{A}) = U_k(\rho_1, \tilde{\rho}_2, \bar{A}) - j_l \sigma_l - j_s \sigma_s - c_A \xi. \tag{14}$$

$U_k$ is the chirally symmetric part of the meson potential. $j_l$ and $j_s$ are explicit chiral symmetry breaking sources that account for the finite current quark masses of the light and the strange quarks. As before, we assume light isospin symmetry. The 't Hooft determinant $\xi = \det(\Sigma) + \det(\Sigma^\dagger)$ effectively incorporates the anomalous breaking of $U(1)_A$ [64–66]. For simplicity, we restrict ourselves to two out of a total of $N_f$ chiral invariants,

$$\rho_1 = \mathrm{tr}\,\Sigma\Sigma^\dagger, \qquad \tilde{\rho}_2 = \mathrm{tr}\Big(\Sigma\Sigma^\dagger - \frac{1}{2}\rho_2 \mathbb{1}\Big)^2. \tag{15}$$

With the total effective potential $V_k = \widetilde{U}_k + U_{\mathrm{glue}}$ and the solution $\bar{\Phi}_k(T, \mu_B, \mu_S)$ of the equations of motion,

$$\left.\frac{\partial V_k(\Phi)}{\partial \Phi}\right|_{\bar{\Phi}_k} = 0, \tag{16}$$

where $\Phi = (\Sigma, \bar{A})$, the $k$-dependent thermodynamic potential is given by

$$\Omega_k = V_k(\bar{\Phi}_k). \tag{17}$$

It can be used to define the cumulants of baryon number and strangeness,

$$\chi_{ij}^{BS} = -T^{i+j-4}\frac{\partial^{i+j}\Omega_0(T, \mu_B, \mu_S)}{\partial\mu_B^i \partial\mu_S^j}. \tag{18}$$

Net baryon number and strangeness are given by the first cumulants, and their densities are obtained by dividing out the spatial volume $\mathcal{V}$,

$$n_B = \frac{\langle N_B - N_{\bar{B}}\rangle}{\mathcal{V}} = \chi_{10}^{BS}\,T^3,$$

$$n_S = \frac{\langle N_{\bar{S}} - N_S\rangle}{\mathcal{V}} = \chi_{01}^{BS}\,T^3. \tag{19}$$

Note that strange antiquarks are defined to have $\langle S\rangle = n_S\mathcal{V} = 1$. In the presence of a large strange chemical potential it might be necessary to take the difference between the light and strange sectors into account also in the symmetric part of the effective potential. This can be achieved by first redefining the generators such that they decompose into purely strange and non-strange parts,

$$\begin{pmatrix}\widetilde{T}^0 \\ \widetilde{T}^8\end{pmatrix} = \frac{1}{\sqrt{3}}\begin{pmatrix}\sqrt{2} & 1 \\ 1 & -\sqrt{2}\end{pmatrix}\begin{pmatrix}T^0 \\ T^8\end{pmatrix}, \tag{20}$$

while keeping

$$\widetilde{T}^{a\in\{1,\dots,7\}} = T^{a\in\{1,\dots,7\}}. \tag{21}$$

Eq. (20) is the rotation from the singlet-octet to the light-strange basis of $U(N_f)$. The respective fields are

$$\Sigma^{(L)} = \widetilde{T}^{a\in\{0,1,2,3\}}(\sigma_a + i\pi_a),$$

$$\Sigma^{(S)} = \widetilde{T}^{a\in\{4,5,6,7,8\}}(\sigma_a + i\pi_a). \tag{22}$$

$\widetilde{T}^{a\in\{0,1,2,3\}}$ are generators of $U(2)$, but embedded in $U(3)$. Since $\Sigma^{(S)}$ contains all generators with non-vanishing off-diagonal entries in the third column and/or row, it contains the open strange mesons, i.e., those with strangeness $S = \pm 1$. With this, the new invariants can straightforwardly be constructed. Note that there are no mixed invariants since $\text{tr}\, T^a T^b = \delta^{ab}/2$. But for now, we will not do this and work with the fully symmetric potential $U_k$. This is a good approximation as long as the strange chemical potential is not too large. For instance, At $T = 0$ and $\mu_B = 0$ one expects kaon condensation if $\mu_S \gtrsim m_K$. In this case, one would certainly have to construct the effective action based on the fields in Eq. (22). But as we discuss below, we are only interested in strange chemical potentials $\mu_S \lesssim 200$ MeV where Eq. (14) is expected to be a good approximation.

## 2.3 Gluonic background

The Euclidean action of $SU(N_c)$ Yang-Mills theory at finite temperature $T$ is invariant under 'twisted' gauge transformations $\mathcal{U}$ which obey for $\beta = 1/T$

$$\mathcal{U}(x_0 + \beta, \vec{x}) = z_n \mathcal{U}(x_0, \vec{x}), \tag{23}$$

where $z_n$ is an element of the center of the gauge group, i.e. $z_n = \mathbb{1}e^{i2\pi n/N_c}$ for $SU(N_c)$. The Polyakov loop [67],

$$L(\vec{x}) = \frac{1}{N_c}\text{tr}\,\mathcal{P}e^{ig\int_0^\beta dx_0 A_0(x_0,\vec{x})}, \tag{24}$$

where $\mathcal{P}$ is the path ordering and the trace is in the fundamental representation, is invariant under gauge transformations but not under center transformations, $L \to z_n L$. The expectation value of the Polyakov loop is related to the free energy $F_{q\bar{q}}$ of a quark-antiquark pair at infinite distance [68],

$$\langle L \rangle \sim e^{-\frac{1}{2}\beta F_{q\bar{q}}}. \tag{25}$$

In (25) we have used declustering and $\langle \bar{L} \rangle = \langle L \rangle$. Confinement implies that it takes an infinite energy to remove the antiquark from the system, and hence $F_{q\bar{q}}$ has to be infinity. Accordingly $\langle L \rangle = 0$. In the deconfined phase the free energy of an isolated quark is finite and thus $\langle L \rangle \neq 0$. Hence, the Polyakov loop serves as an order parameter for the deconfinement transition in the static limit, which can be associated to the breaking/restoration of center symmetry.

In the spirit of the present mean-field theory for gluons the Polyakov loop is taken into account by a temporal gluonic background $\bar{A}_\mu = \delta_{\mu 0}\bar{A}_0$, as already mentioned before. As the effective action is invariant under background gauge transformations, the (constant)

background gauge field can be rotated into the Cartan subalgebra, to wit,

$$\frac{g}{2\pi T}\bar{A}_0 = \frac{g}{2\pi T}\left(\bar{A}_0^{(3)}t^3 + \bar{A}_0^{(8)}t^8\right) \tag{26}$$

$$= \frac{\varphi_3}{2}\begin{pmatrix} 1 & 0 & 0 \\ 0 & -1 & 0 \\ 0 & 0 & 0 \end{pmatrix} + \frac{\varphi_8}{2\sqrt{3}}\begin{pmatrix} 1 & 0 & 0 \\ 0 & 1 & 0 \\ 0 & 0 & -2 \end{pmatrix},$$

where we defined

$$\varphi_i = \frac{g\bar{A}_0^{(i)}}{2\pi T}, \qquad i = 3, 8, \tag{27}$$

for the eigenvalues of the temporal gauge field. Inserting this into Eq. (24), the integral and trace become trivial and the Polyakov loop and antiloop are:

$$L = \frac{1}{3}e^{i\frac{\pi}{\sqrt{3}}\varphi_8}\left(e^{-i\sqrt{3}\pi\varphi_8} + 2\cos(\pi\varphi_3)\right), \tag{28}$$

$$\bar{L} = \frac{1}{3}e^{-i\frac{\pi}{\sqrt{3}}\varphi_8}\left(e^{i\sqrt{3}\pi\varphi_8} + 2\cos(\pi\varphi_3)\right). \tag{29}$$

Since we are working in a field theoretical approach with a gauge field $A_\mu$ we should use $L[\langle A_0\rangle] = L[\bar{A}_0]$, instead of $\langle L[A_0]\rangle$ as computed on the lattice [69, 70]. The former variable shows a more rapid transition from the confined to the deconfined phase, and is saturated by unity for temperatures $T \gtrsim 1.25\, T_c$. The difference is accounted for with a trivial, but temperature-dependent normalisation factor, for more details see [71]. In the present work we use a mean field approximation for the glue dynamics leading to $L[\langle A_0\rangle] = \langle L[A_0]\rangle$. This approximation will be lifted in future work.

Note also that our effective action (13) is manifestly gauge invariant since the gluon background field only appears in the covariant derivative of the quarks and the gauge invariant Polyakov loops, which are the variables of the gluon effective potential as discussed below.

The idea underlying the above formulation has been proven to be very successful in Matrix- or Polyakov-loop models, where the simple representation of the gluon field in (26) leads to particularly simple expression of $L$, while still being able to capture main features of confinement, see e.g. [34] and references therein. By now this has been also worked out for full QCD [69, 71, 72], which provides a natural embedding of the current model into QCD as a QCD-assisted effective field theory, e.g. [73].

At finite chemical potential another intricacy has to be taken care of: since quarks and antiquarks manifest themselves in the effective action with terms

$$L\, e^{-\mu_q/T}, \quad \text{and} \quad \bar{L}\, e^{\mu_q/T}, \tag{30}$$

in the fermion occupation numbers, they have to be real valued in order to give a well-defined equation of state. Here, we defined the quark chemical potential $\mu_q = \mu_B/3$. Furthermore, at finite chemical potential they are also unequal. Hence, while one can assume without loss of generality that $\varphi_8 = 0$ at $\mu = 0$, it has to be non-zero and imaginary at finite $\mu$,

$$\bar{\varphi}_8 = -i\varphi_8, \quad \bar{\varphi}_8 \in \mathbb{R}. \tag{31}$$

The loops then are

$$L = \frac{1}{3}e^{-\frac{\pi}{\sqrt{3}}\bar{\varphi}_8}\left(e^{\sqrt{3}\pi\bar{\varphi}_8} + 2\cos(\pi\varphi_3)\right),$$

$$\bar{L} = \frac{1}{3}e^{\frac{\pi}{\sqrt{3}}\bar{\varphi}_8}\left(e^{-\sqrt{3}\pi\bar{\varphi}_8} + 2\cos(\pi\varphi_3)\right). \tag{32}$$

This was pointed out, e.g., in [33, 74–76]. In practice, the transition from a QM to a PQM model can be achieved by a simple replacement of the quark distribution function, $n_F \to N_F$, in many cases. The reason is that the $\bar{A}_0$ eigenvalues enter the computation as a $SU(N_c)$-valued imaginary shift of the chemical potential, cf. Eq. (6). Hence, in any finite-temperature loop computation where the chemical potential only enters through the Fermi-Dirac distribution, the non-trivial color trace (i.e. the sum over the eigenvalues) simply results in a modified distribution function,

$$N_F(E_q, \mu_q; L, \bar{L}) = \frac{1 + 2\bar{L}e^{(E_q - \mu_q)/T} + Le^{2(E_q - \mu_q)/T}}{1 + 3\bar{L}e^{(E_q - \mu_q)/T} + 3Le^{2(E_q - \mu_q)/T} + e^{3(E_q - \mu_q)/T}}, \tag{33}$$

where $E_q$ is the quark quasiparticle energy, $E_q = \sqrt{k^2 + m_q^2}$, where $k$ is the modulus of the spatial momentum. But note that we pointed out in [45] that this simple replacement is not always correct. The modified distribution function has a very useful qualitative interpretation: in the confined phase with $L \approx 0$ one has $N_F \approx 1/\{\exp[3(E_q - \mu_q)/T] + 1\}$, which is the distribution function for a $qqq$-state, a baryon. See [43] for a more careful discussion of this behavior. In the deconfined phase $N_F$ is identical to the distribution of a single quark. The terms $\exp[2(E_q - \mu_q)/T]$ in Eq. (33) can be interpreted as intermediate diquark states. So the coupling of the gluon background field $\bar{A}_0$ to the quarks leads to a smooth interpolation between baryons in the hadronic phase and quarks in the QGP. Even though the effective action in Eq. (13) only has mesons as explicit hadronic content, we can still account for baryon dynamics. Including both a baryon- and a strange chemical potential allows us to capture the effects of strange and nonstrange baryons separately.

To be able to capture the deconfinement phase transition, an effective gluon potential is necessary. The strategy for Polyakov-loop enhanced models is to use a phenomenological parametrization of the effective potential of the pure gauge theory at finite temperature in terms of Polyakov loops. In this work we use the parametrization introduced in [77] with $U_{\text{glue}}(\bar{A}) = U_{\text{glue}}(L, \bar{L})$ given by

$$\frac{U_{\text{glue}}(L, \bar{L})}{T^4} = -\frac{1}{2}a(T)\bar{L}L + b(T)\ln\left[M_H(L, \bar{L})\right] + \frac{1}{2}c(T)(L^3 + \bar{L}^3) + d(T)(\bar{L}L)^2, \tag{34}$$

where $M_H$ is the $SU(3)$ Haar measure in terms of the Polyakov loops,

$$M_H(L, \bar{L}) = 1 - 6\bar{L}L + 4(L^3 + \bar{L}^3) - 3(\bar{L}L)^2. \tag{35}$$

The advantage of this parametrization is that it reproduces the pressure and the Polyakov loop susceptibilities of $SU(3)$ Yang-Mills theory. The relevance of an accurate description of Polyakov loop susceptibilities in particular for the cumulants of particle number distributions has been discussed in [34, 43] and explicitly demonstrated in [45]. The explicit choice for the parameters $a$, $b$ and $c$ is discussed in App. A. There, we also discuss how the chemical potential dependence of the Polyakov loop potential is modelled.

By relying on a parametrization of the gauge potential based only on Yang-Mills theory, we make sure that all effects related to matter fluctuations, i.e. the unquenching, are included dynamically within our model through the coupling of $\bar{A}_0$ to the quarks. Since this is not put in by hand here, it adds to the predictive power of the model.

# 3 Fluctuations

It has been shown that even for zero chemical potentials at the very least pion fluctuations are required to get reasonably accurate results for the QCD equation of state [51]. We argued

that for strangeness dynamics kaons are the most relevant degrees of freedom at small and moderate chemical potentials as they are the lightest strange particles in the hadronic sector. So without kaon fluctuations crucial effects related to finite $\mu_S$ would certainly be missed. To account for meson fluctuations we use the functional renormalization group. It is a semi-analytical method providing a non-perturbative regularization and renormalization scheme for the resummation of an infinite class of Feynman diagrams. For reviews of the FRG we refer the reader to [54, 78–83].

## 3.1 The Functional Renormalization Group

The FRG realizes Wilson's renormalization group idea of successively integrating out quantum fluctuations from large to small energy scales. The starting point is the microscopic action $\Gamma_{k=\Lambda}$ at some large initial momentum scale $\Lambda$ in the UV. By lowering the RG scale $k$, quantum fluctuations are successively integrated out until one arrives at the full macroscopic quantum effective action $\Gamma \equiv \Gamma_{k=0}$ at $k = 0$. Ideally, one starts in the perturbative regime where the initial effective action $\Gamma_{k=\Lambda}$ is related to the well-known microscopic action of QCD. As already discussed before, in the present low-energy approach we choose $\Lambda$ at a scale where we assume that gluon degrees of freedom are already integrated out. Hence, $\Lambda$ is directly linked to the Yang-Mills mass gap with $\Lambda \lesssim 1\,\text{GeV}$. In Landau gauge QCD the Yang-Mills mass gap is reflected in the gapping of the gluon propagator which leads to an effective suppression of gluonic diagrams in a functional approach such as the FRG, see the reviews [16–19] and references therein.

The FRG formulates the RG in terms of a functional differential equation for the evolution of the scale dependent effective action $\Gamma_k$, the Wetterich equation [84–86]. In the present case, with dynamical quarks and mesons in a gluon background, the flow equation reads

$$\partial_t \Gamma_k = \frac{1}{2} \sum_{i=1}^{2N_f^2} \text{Tr}\left(G_{\phi_i \phi_i, k} \cdot \partial_t R_k^{\phi_i}\right) - 2\text{Tr}\left(G_{l\bar{l}, k} \cdot \partial_t R_k^l\right) - \text{Tr}\left(G_{s\bar{s}, k} \cdot \partial_t R_k^s\right), \qquad (36)$$

where $\partial_t = k\frac{d}{dk}$ denotes the logarithmic scale derivative. The trace runs over all discrete and continuous indices, i.e. color, spinor and the loop momenta and/or frequencies respectively. The sum in the first line is over all $2N_f^2$ scalar and pseudoscalar mesons in Eq. (8). The generalized meson and quark propagators are given by matrix elements in field space,

$$G_{\Phi_i \Phi_j, k}[\Phi] = \left[\frac{1}{\Gamma_k^{(2)}[\Phi] + R_k}\right]_{\Phi_i \Phi_j} (p, -p), \qquad (37)$$

with the generalized field $\Phi = (\phi, q, \bar{q}, \bar{A}_0)$, $R_k$ is the matrix of regulators $R_k^{\phi_i}, R_k^l, R_k^s$ being diagonal for the mesons and symplectic for the quarks, and $\Gamma_k^{(2)} = \delta^2 \Gamma_k / \delta \Phi^2$. Since we assume isospin symmetry we define the light quark as $l \equiv u = d$ and the quark field becomes $q = (l, l, s)$. The scale-dependent IR regulators $R_k^{\Phi_i}$ can be understood as momentum-dependent masses that suppress the infrared modes of the field $\Phi_i$. In addition, the terms $\partial_t R_k^{\Phi_i}$ in Eq. (36) also ensure UV-regularity. Their definitions and a more explicit form of the flow equation will be discussed in the next section. We use the local potential approximation (LPA) here, which means that only the symmetric part of the meson effective potential, $U_k$, is running in Eq. (13). For a study of effects beyond LPA in the QM at finite temperature and density we refer to [42, 49]. While effects beyond LPA are certainly relevant, at least the qualitative features of the relevant physics for the present purposes are captured by the running of the effective potential.

The FRG is a method to integrate out quantum fluctuations in Euclidean spacetime in terms of one-particle irreducible (1PI) diagrams. Consequently, the dynamics is driven by

quantum fields propagating as internal lines of 1PI Feynman diagrams with Euclidean momenta. All interactions are governed by off-shell fields. This implies a very simple hierarchy for dynamically relevant contributions: the lighter the degree of freedom, the more relevant it is. This means in particular that the contribution of particles with masses $m \gtrsim \Lambda$ to, for instance, the equation of state, is negligible. Within this fluctuation-driven approach one therefore expects that kaons and $s$-quarks coupled to $\bar{A}_0$ are sufficient to capture the relevant strangeness effects at small to moderate chemical potentials in the same way that the dynamics of pions and quarks coupled to $\bar{A}_0$ already give almost quantitative results for the equation of state at vanishing chemical potentials, cf. [51]. This is in contrast to purely statistical approaches without quantum fluctuations, such as the HRG [1], where the lack of dynamics and interactions has to be compensated by taking into account all possible hadrons and their excited states. While being very successful in the description of particle properties at the freeze-out, the QCD phase transition and features of the QGP are not accessible in such approaches.

## 3.2 Flow of the effective potential

Here, we briefly discuss the RG flow equations of our model. For $\mu_S = 0$ this has been discussed in [48–51]. We therefore focus on the manifestly new contributions to the flow equation here. As discussed in Sec. 2.1, the non-vanishing strange chemical potential also couples to the open strange mesons. In our case these are the four scalar kappa-mesons and the four pseudoscalar kaons. Induced by the covariant derivative $\bar{D}_\nu$ in Eq. (9), this leads to a shift of the frequency in the kinetic terms of these particles. All other mesons are unaffected by finite $\mu_S$. Their contributions to the flow of the effective potential is therefore identical to the ones in, e.g., [49]. We will therefore only outline the changes for the open strange mesons. For definiteness, we pick out the contribution of the charged kaons, $K^\pm$. Within the present approximation, the regulated propagator defined in Eq. (37) is:

$$G_{K^+K^-,k}(p_0, \vec{p}; \mu_S) = \frac{1}{(p_0 - i\mu_S)^2 + \vec{p}^{\,2}\big(1 + r_B(\vec{p}^{\,2})\big) + m_{K,k}^2}, \tag{38}$$

where the delta distribution for momentum conservation is omitted. Note that finite $\mu_S$ leads to a linear frequency term in the propagator. We choose to regulate only the spatial momenta with a regulator of the form $R_k^\phi = \vec{p}^{\,2} r_B(\vec{p})$. Nonetheless, both UV and IR regularity for arbitrary frequencies is still guaranteed. We use the flat or Litim regulator with the shape function $r_B(\vec{p}^{\,2}) = (k^2/\vec{p}^{\,2} - 1)\Theta(k^2 - \vec{p}^{\,2})$ [87, 88]. For the antiparticle propagator, only the sign of $\mu_S$ changes,

$$G_{K^-K^+,k}(p_0, \vec{p}; \mu_S) = G_{K^+K^-,k}(p_0, \vec{p}; -\mu_S). \tag{39}$$

Inserting this into the flow equation (36), we find

$$\frac{1}{2}\mathrm{STr}\, G_{K^+K^-,k}\, \partial_t R_k^K$$

$$= \frac{1}{2} T \sum_{n\in\mathbb{Z}} \int \frac{d^3p}{(2\pi)^3} G_{K^+K^-,k}(\omega_n, \vec{p}, \mu_S)\, \partial_t R_k^K(\vec{p})$$

$$= \frac{k^4}{12\pi^2} \frac{k}{E_K} \Big[ 1 + n_B(E_K - \mu_S) + n_B(E_K + \mu_S) \Big]$$

$$\equiv \frac{k^4}{4\pi^2} \bar{l}_0^{(K)}(\mu_S), \tag{40}$$

where $\omega_n = 2\pi nT$ is the bosonic Matsubara frequency, $n_B(E) = [\exp(E/T) - 1]^{-1}$ is the Bose-Einstein distribution, $E_K = \sqrt{k^2 + m_{K,k}^2}$ is kaon energy. In this form the thermal particle, antiparticle as well as the vacuum contribution of open strange mesons are manifest. Since this expression is symmetric under exchange of particles and antiparticles ($\mu_S \to -\mu_S$), it also holds for the $K^- K^+$-contributions as well as for $K^0$ and $\bar{K}^0$. For the contribution of the $\kappa$'s, only the quasiparticle energy has to be replaced, $E_K \to E_\kappa$.

The flow of the effective potential in terms of the physical fields is given by

$$\partial_t U_k(\rho_1, \tilde{\rho}_2) =$$

$$\frac{k^4}{4\pi^2} \Big\{ \bar{l}_0^{(f_0)}(0) + 3\bar{l}_0^{(a_0)}(0) + 4\bar{l}_0^{(\kappa)}(\mu_S) + \bar{l}_0^{(\sigma)}(0)$$

$$+ \bar{l}_0^{(\eta)}(0) + 3\bar{l}_0^{(\pi)}(0) + 4\bar{l}_0^{(K)}(\mu_S) + \bar{l}_0^{(\eta')}(0)$$

$$- 4N_c \Big[ 2\bar{l}_0^{(l)}(\mu_q) + \bar{l}_0^{(s)}(\mu_q - \mu_S) \Big] \Big\}, \tag{41}$$

with the quark threshold function

$$\bar{l}_0^{(q)}(\mu) = \frac{k}{3E_q} \Big[ 1 - N_F(E_q, \mu; L, \bar{L}) + \bar{N}_F(E_q, \mu; L, \bar{L}) \Big], \tag{42}$$

and the Fermi-Dirac distribution in presence of a non-vanishing $A_0$ background $N_F$ (33). The antiquark distribution function is given by $\bar{N}_F(E_l, \mu; L, \bar{L}) = N_F(E_l, -\mu; \bar{L}, L)$. Eq. (41) is identical to the one used in [49], except that $\mu_S$ now enters the threshold functions of the open strange mesons through the distribution function in Eq. (40).

## 3.3 Flow of the particle numbers

The computation of the cumulants of particle number distributions require derivatives of the thermodynamic potential with respect to the chemical potential, cf. Eq. (18). While it is simple to perform these derivatives numerically, many points in $\mu_{B,S}$ are required to ensure numerical accuracy and for higher cumulants this is practically not feasible. One alternative is to use algorithmic derivation techniques, see e.g. [89]. The other alternative is given by solving the flow equations for the cumulants directly. For first discussions in this direction we refer to [43, 46]. In both cases, the accuracy of a cumulant of arbitrary order is given by the accuracy of the differential equation solver that is used and numerical derivatives on the data are obsolete. We will not give an exhaustive discussion here and restrict ourselves to the cases directly relevant for the present work.

It is straightforward to derive flow equations for the cumulants. For the first cumulants, i.e. the particle numbers, this is particularly simple due to

$$\frac{d\Omega_k}{d\mu} = \frac{\partial \Omega_k}{\partial \mu} + \frac{\partial \Omega_k}{\partial \Phi} \frac{\partial \Phi}{\partial \mu} = \frac{\partial \Omega_k}{\partial \mu}, \tag{43}$$

where $\Phi$ contains all meson and quark fields as well as the Polyakov loop and antiloop. In the last step, the equations of motion were used. Hence, only the explicit dependence of the effective potential on $\mu$ is relevant here. Within the LPA we use in the present work, only the effective potential is running and, under the assumption that one can interchange the RG scale derivative and the $\mu$-derivative, a simple flow equation for the strangeness number density $n_S$ is obtained,

$$\partial_t n_{S,k} = -\frac{k^4}{\pi^2} \Big[ \partial_{\mu_S} \bar{l}_0^{(\kappa)}(\mu_S) + \partial_{\mu_S} \bar{l}_0^{(K)}(\mu_S) - N_c \partial_{\mu_S} \bar{l}_0^{(s)}(\mu_q - \mu_S) \Big]. \tag{44}$$

As discussed above and in App. A, the Polyakov loop potential $U_{\mathrm{glue}}$ also carries an explicit $\mu_S$ dependence. Since $U_{\mathrm{glue}}$ does not run, we can store its contribution into the initial condition for convenience. If the initial action would be $\mu_S$-independent, the initial strangeness would then be trivially given by $n_{S,\Lambda} = -\partial_{\mu_S}\Omega_\Lambda = -\partial_{\mu_S}U_{\mathrm{glue}}$. However, as we discuss in the next section, there is an important in-medium correction to the initial potential, $\Delta\Gamma_\Lambda$, so we provide the explicit equation for the initial strangeness number in the next section.

Since the mesons do not carry baryon number, the flow equation for the corresponding density is just given by the fermion contribution,

$$\partial_t n_{B,k} = \frac{N_c k^4}{\pi^2}\partial_{\mu_B}\Big[2\bar{l}_0^{(l)}(\mu_q) + \bar{l}_0^{(s)}(\mu_q - \mu_S)\Big]. \tag{45}$$

Again we store the $k$-independent gluon contribution in the initial conditions. This will be discussed in the next section.

## 4 Results

### 4.1 Initial Conditions

The scale set by temperatures above the critical temperature $T_c$ exceeds the cutoff scale $\Lambda$ of the effective model, $2\pi T \gtrsim \Lambda$. In order to describe thermodynamic quantities above $T_c$, we therefore need initial conditions that depend on the temperature and, since we are interested in finite chemical potential effects as well, also on $\mu$. These initial conditions are governed by integrating out fluctuations from scales $\bar\Lambda \gg 2\pi T$ down to $\Lambda$. Hence, we want to correct our vacuum initial conditions for in-medium effects at the initial scale, for a recent detailed discussion see [90]. This is achieved by integrating the initial vacuum effective action from $\Lambda$ to $\bar\Lambda$ and subsequently integrating the in-medium effective action down to $\Lambda$ again [91],

$$\Delta\Gamma_\Lambda(T,\mu_q,\mu_S) = \int_\Lambda^{\bar\Lambda}\frac{dk}{k}\Big[\partial_t\Gamma_k(0,0,0) - \partial_t\Gamma_k(T,\mu_q,\mu_S)\Big]. \tag{46}$$

As long as the scale set by the medium parameters is smaller than $\Lambda$, $\Delta\Gamma_\Lambda(T,\mu_q,\mu_S)$ vanishes because the in-medium flow and the vacuum flow are identical for $k \geq \Lambda$. Since quark fluctuations certainly dominate over meson fluctuations for $\Lambda \gtrsim 900\,\mathrm{MeV}$, we can approximate the flows in Eq. (46) by the purely fermionic ones, to wit,

$$\Delta\Gamma_\Lambda(T,\mu_q,\mu_S) = -\int_\Lambda^\infty dk \frac{N_c k^4}{3\pi^2}\bigg\{\frac{2}{E_l}\Big[N_F(E_l,\mu_q;L,\bar{L})$$

$$+ \bar{N}_F(E_l,\mu_q;L,\bar{L})\Big] + \frac{1}{E_s}\Big[N_F(E_s,\mu_q-\mu_S;L,\bar{L})$$

$$+ \bar{N}_F(E_s,\mu_q-\mu_S;L,\bar{L})\Big]\bigg\}. \tag{47}$$

We set $\bar\Lambda \to \infty$ since the thermal contribution to the quark flow is UV regular.

It is important to note that $\Delta\Gamma_\Lambda$ not only depends on the medium parameters but also on the field expectation values. The dependence on the gluon background field in the current mean field approximation for the glue dynamics enters through the Polyakov loops $L, \bar{L} = \langle L[A_0]\rangle, \langle \bar{L}[A_0]\rangle$, and the meson field expectation values through the quark masses. Since the Polyakov loop expectation values approach their deconfined value only for $T \gtrsim 4T_c$, cf. [94], non-trivial values for $L, \bar{L}$ have to be taken into account in Eq. (47). Note that this may

Table 1: Parameters for the initial effective action and the Polyakov loop potential. They are chosen such that we find in the vacuum at $k = 0$ for the pion mass $m_\pi = 138$ MeV, for the kaon mass $m_K = 495$ MeV, for the $\sigma$-meson mass $m_\sigma = 463$ MeV, for the sum $m_\eta^2 + m_{\eta'}^2 = 1.218$ GeV$^2$, for the light current quark mass $m_l = 302$ MeV and for the decay constants $f_\pi = 93$ MeV and $f_K = 113$ MeV. The last two parameters belong to the Polyakov loop potential and are fixed by the pressure of 2+1 flavor lattice QCD at vanishing chemical potentials, see App. A.

| parameter | value |
|---|---|
| $\Lambda$ | $0.9$ GeV |
| $\lambda_{10,\Lambda}$ | $(0.830 \, \text{GeV})^2$ |
| $\lambda_{20,\Lambda}$ | $10$ |
| $\lambda_{01,\Lambda}$ | $54$ |
| $h$ | $6.5$ |
| $j_l$ | $(0.121 \, \text{GeV})^3$ |
| $j_s$ | $(0.336 \, \text{GeV})^3$ |
| $c_A$ | $4.808$ GeV |
| $b_0$ | $1.6$ |
| $\alpha_t$ | $0.47$ |

change when going beyond the mean field approximation for the glue dynamics. As discussed before, $L[\langle A_0 \rangle]$ approaches unity far more rapidly [71].

Furthermore, if the meson part of the effective potential is computed away from its stationary point, the relevant quark masses are those given by $m_l = h\sigma_l/2$ and $m_s = h\sigma_s/\sqrt{2}$, where $\sigma_l$ and $\sigma_s$ are the meson background fields which, in general, do not have to coincide with their vacuum expectation values as long as one is still able to reliably solve the corresponding equation of motion for the mesons (e.g. by sampling the potential on a grid of field configurations as in [48, 50] or by using the fixed background Taylor expansion as in [42, 49]). With all the background- and medium-dependencies spelled out explicitly, the initial potential is

$$\Omega_\Lambda(\sigma_l, \sigma_s, L, \bar{L}; T, \mu_B, \mu_S) = \widetilde{U}_\Lambda(\sigma_l, \sigma_s) + \Delta\Gamma_\Lambda(\sigma_l, \sigma_s, L, \bar{L}; T, \mu_B, \mu_S)$$
$$+ U_{\text{glue}}(L, \bar{L}; T, \mu_B, \mu_S), \tag{48}$$

where we added the $U_{\text{glue}}$ for convenience. Since it does not depend on the RG scale $k$, it is irrelevant whether we add it to the initial or to the final potential. Since it also carries no dependence on the meson fields, it only contributes to the pressure and leaves the initial meson $n$-point functions unaffected. The initial meson potential is

$$\widetilde{U}_\Lambda(\sigma_l, \sigma_s) = U_\Lambda(\rho_1, \tilde{\rho}_2) - j_l\sigma_l - j_s\sigma_s - c_A\frac{\sigma_l^2\sigma_s}{2\sqrt{2}}$$

$$= \lambda_{10,\Lambda}\rho_1 + \frac{1}{2}\lambda_{20,\Lambda}\rho_1^2 + \lambda_{01,\Lambda}\tilde{\rho}_2 \tag{49}$$

$$- j_l\sigma_l - j_s\sigma_s - c_A\frac{\sigma_l^2\sigma_s}{2\sqrt{2}}.$$

It is sufficient to take only relevant and marginal operators into account at the initial scale since meson fluctuations are small at high energies and irrelevant operators are dimensionally

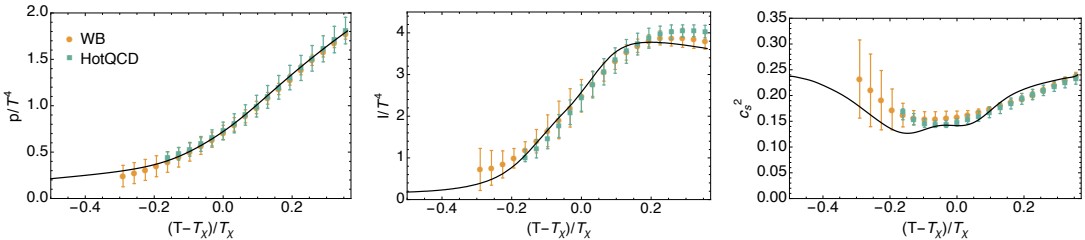

Figure 1: The pressure $p$, the trace anomaly $I$ and the speed of sound squared $c_s^2$ at $\mu_B = \mu_S = 0$ in comparison to lattice results. The temperature has been rescaled to $t \equiv (T - T_\chi)/T_\chi$ du to different pseudocritical temperatures in our model and on the lattice. The HotQCD collaboration data is from [92] and the Wuppertal-Budapest collaboration (WB) data from [93].

suppressed in addition. Note that irrelevant operators are generated by the RG flow at smaller scales and are quantitatively and qualitatively relevant [42]. Our initial values are listed in Tab. 1. The last two parameters are free parameters of the Polyakov loop potential and are discussed in App. A. In general, these initial parameters have uncertainties related to the uncertainties in the masses and decay constants we use to fix them. However, these uncertainties are irrelevant within the scope of the present work and are therefore neglected.

The total contribution to the initial conditions for mesonic $n$-point functions can be expanded as:

$$U_\Lambda(\rho_1, \tilde{\rho}_2) + \Delta\Gamma_\Lambda(\sigma_l, \sigma_s, L, \bar{L}; T, \mu_B, \mu_S) = \sum_{n,m=0}^{N} \frac{\omega_{nm,\Lambda}}{n!m!}(\rho_1 - \kappa_1)^n(\tilde{\rho}_2 - \kappa_2)^m, \qquad (50)$$

and as a consequence of the discussion above the expansion coefficients are

$$\omega_{nm,\Lambda} = \lambda_{nm,\Lambda} + \frac{\partial^{n+m}\Delta\Gamma_\Lambda}{\partial\rho_1^n \partial\tilde{\rho}_2^m}\bigg|_{\kappa_1,\kappa_2}. \qquad (51)$$

Following Eq. (49) only the renormalizable initial parameters of the chirally symmetric part of the effective potential, $\lambda_{10,\Lambda}$, $\lambda_{20,\Lambda}$, $\lambda_{01,\Lambda}$, are nonzero. However, due to the meson background field dependence of $\Delta\Gamma_\Lambda$, these and higher order initial couplings receive medium- and gluon background dependent corrections. As the explicit symmetry breaking parameters $j_l$, $j_s$ and $c_A$ do not run within the present approximation, they are unaffected. We discuss viable simplifications of these complicated initial conditions in App. C.

The initial conditions for flows of the particle numbers are also affected by $\Delta\Gamma_\Lambda$. As discussed in the previous section, we store the contribution of the glue potential in the initial conditions for convenience. Thus, we find for the the strangeness and baryon number densities:

$$n_{S,\Lambda} = -\partial_{\mu_S}\Delta\Gamma_\Lambda - \partial_{\mu_S}U_{\text{glue}},$$
$$n_{B,\Lambda} = -\partial_{\mu_B}\Delta\Gamma_\Lambda - \partial_{\mu_B}U_{\text{glue}}. \qquad (52)$$

The system of flow equation is solved by using the fixed background Taylor expansion developed in [42, 49].

## 4.2 Comparison to lattice gauge theory

To demonstrate the validity of our model at vanishing chemical potentials, we compare our results on thermodynamic quantities to the results of lattice gauge theory. Within our model,

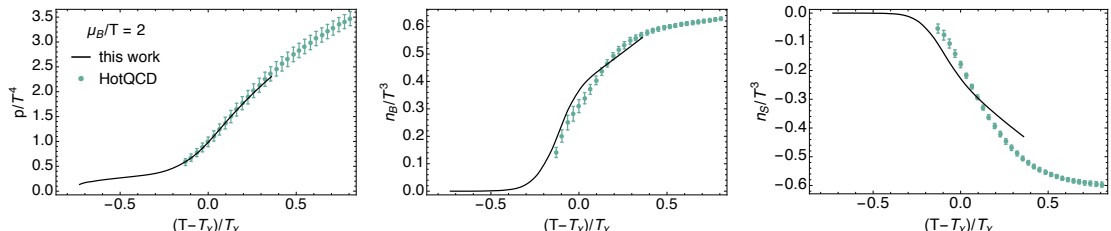

Figure 2: The pressure $p$, the baryon number density $n_B$ and the strangeness density $n_S$ at $\mu_B/T = 2$ and $\mu_S = 0$ in comparison to lattice results as a function of the rescaled temperature $t$. The lattice data is taken from [8].

the pseudocritical temperature of the chiral transition, which we define as the location of the inflection point of the subtracted chiral condensate,

$$\Delta_{LS} = \frac{\left(\sigma_L - \frac{j_L}{j_S}\sigma_S\right)\big|_T}{\left(\sigma_L - \frac{j_L}{j_S}\sigma_S\right)\big|_{T=0}}, \tag{53}$$

is $T_\chi = 176.5$ MeV. This is roughly 15% larger than the pseudocritical temperature found on the lattice [95] so the absolute scale in our computation differs from the lattice. We therefore use relative temperature scales $t = (T - T_\chi)/T_\chi$ for our comparison. This allows us to compare the overall shapes of the functions which are sensitive to the relevant dynamics. The pressure, $p$, entropy density, $s$, energy density, $\epsilon$, trace anomaly, $I$, and the speed of sound squared, $\tilde{c}_s^2$, are defined as follows,

$$p = -\Omega_0,$$

$$s = \frac{\partial p}{\partial T},$$

$$\epsilon = -p + Ts + \mu_B n_B + \mu_S n_S, \tag{54}$$

$$I = \epsilon - 3p,$$

$$\tilde{c}_s^2 = \frac{s(T, \mu_B, \mu_S)}{\partial \epsilon(T, \mu_B, \mu_S)/\partial T}.$$

We note that, strictly speaking, $\tilde{c}_s^2$ is identical to the (hydrodynamic) speed of sound only at vanishing density, since the latter is defined at fixed entropy and particle numbers. We use $\tilde{c}_s^2$ since we want to compare different thermodynamic quantities for fixed values of the baryon chemical potential and different strangeness chemical potentials.

Our results on the pressure, the interaction measure and the speed of sound squared in comparison to the lattice are shown in Fig. 1. We single out the trace anomaly and the speed of sound since they are sensitive to the particle number densities and to temperature derivatives of the pressure. We find excellent agreement with lattice results for the pressure and the trace anomaly and good agreement for the speed of sound. But note that the former has been used to fix the free parameters of the Polyakov loop potential, cf. the last two parameters in Tab. 1. The speed of sound squared is thermodynamically highly nontrivial since it involves two $T$ derivatives of the pressure. Furthermore, since it is a ratio of two extensive thermodynamic quantities (the entropy and the heat capacity) that grow with the number of degrees of freedom, this effect, which dominates in particular the behavior of the pressure at large $T$, is cancelled to some extent. The two minima of $c_s^2$ in our computation are due to the fact that we find quite

different pseudocritical temperatures of deconfinement, $T_d$, and the chiral transition, with $T_d \approx 155$ MeV if defined as the inflection point of $L(T)$. The first minimum $c_s^2$ then corresponds to the deconfinement transition and the second to the chiral transition.

To check the validity of our simple model also at finite $\mu_B$ we compare it to lattice results obtained from a Taylor expansion of the thermodynamic potential for various $\mu_B/T$ at $\mu_S = 0$ [8]. Fig. 2 shows the results for $\mu_B/T = 2$. We note that the comparison does not change qualitatively for other ratios. Only the temperature is rescaled for comparison but we assumed that the ratio $\mu_B/T$ is the same for our calculation and the lattice. This means that for instance at $t \approx 0.35$ we have $\mu_B = 480$ MeV in our calculation and $\mu_B = 420$ MeV in the lattice results. We have chosen the chiral transition temperatures $T_\chi$ for $\mu_B = 0$ for the definition of $t$. With this, the pressure shows perfect agreement with the lattice even at finite $\mu_B$. The same is true for the entropy density not shown here.

Most sensitive to the finite-$\mu_B$ effects are certainly the particle numbers, since they are only generated by finite chemical potentials in the first place. We therefore also compare our results on $n_B$ and $n_S$ to the lattice results in Fig. 2. The baryon number density agrees very well with the lattice results at $\mu_B/T = 2$. In contrast to the lattice, we see a larger bump in the vicinity of the phase transition. Note that the bump appears in the lattice data only at the highest order in the expansion of the thermodynamic potential presently available, which is $\mu_B^6$ [8]. The error on the lattice data stems from the determination of the expansion coefficients for a given order. The systematic error, e.g., from missing higher-order corrections of the expansion, is unknown. So it is possible that the bump becomes more pronounced in the lattice data at higher orders of the expansion. The strangeness density drops less steep with $t$ in our results, but the overall agreement is still good. We want to emphasize that the difference between $n_B$ and $-n_S$ in our computation stems solely from the fluctuations of open strange mesons at $\mu_S = 0$. So within a mean-field study of the (P)QM/(P)NJL models the physical difference between $n_B$ and $-n_S$ in the hadronic phase at vanishing $\mu_S$ cannot be captured.

The discrepancy between our results and the lattice results for $n_S$ at larger $t$ could be a hint that strange baryon dynamics are not captured quantitatively in the PQM model. As discussed in Sec. 2.3, they enter indirectly through the coupling to the gluon background field. This appears to work very well for $n_B$, on the other hand, indicating that nucleon effects are described well. The three-quark states that contribute through the modified fermion distribution function in Eq. (33) always contain the same quark flavor, so while $lll$-states such as the nucleons or $sss$-states such as the $\Omega$ are effectively taken into account, the dynamically most relevant strange baryons, the $lls$-states $\Lambda$ and $\Sigma$, but also $lss$-states such as the $\Xi$ might not be captured accurately here. This could, rather heuristically, explain the very good agreement of $n_B$ and the small deviations of $n_S$.

### 4.3 $\mu_S$ at strangeness neutrality

We computed the strangeness density $n_S(T, \mu_B, \mu_S)$ for $T \in \{20, \dots, 250\}$ MeV and $\mu_B, 3\mu_S \in \{0, \dots, 675\}$ MeV. We note that the low-energy effective theory is only valid up to moderate chemical potentials so we refrain from exploring the region beyond 675 MeV. This is discussed in detail in App. B. An example of $n_S$ as a function of $\mu_S$ for fixed $\mu_B$ and different $T$ is given in Fig. 3. It is interesting to observe that $n_S$ is a linear function of $\mu_S$ at larger temperatures. The larger $\mu_B$, the smaller the temperature where this linear behavior emerges. Given that $n_S/T^3 = \chi_{01}^{BS}$, we conclude that higher strangeness cumulants $\chi_{0n}^{BS}$ for $n \geq 3$ are highly suppressed at moderate to large temperatures.

The zero crossing of $n_S$ gives the value of $\mu_S$ that enforces strangeness neutrality for given

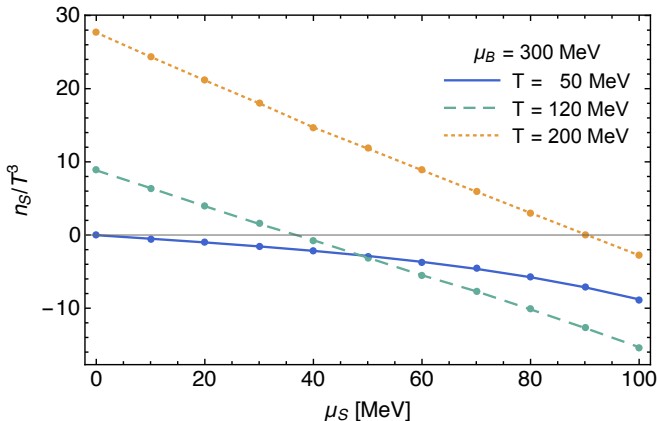

Figure 3: Strangeness density as a function of $\mu_S$ at $\mu_B = 300$ MeV for various temperatures.

$T$ and $\mu_B$. Put differently, $n_S = 0$ implicitly defines the function

$$\mu_{S0}(T, \mu_B) = \mu_S(T, \mu_B)\Big|_{n_S=0}. \tag{55}$$

In Fig. 4 we show our results of $\mu_{S0}$ as a function of $T$ for various $\mu_B$ at strangeness neutrality. We see that it is always a monotonously increasing function of $T$ for the baryon chemical potentials considered here. At large temperatures we find $\mu_{S0} \approx \mu_B/3$, as indicated by the dashed lines at the right edge of the figure. Furthermore, at small temperatures, $T \approx 50$ MeV, $\mu_{S0}$ becomes nonzero only for $\mu_B \gtrsim 400$ MeV. For $\mu_B = 0$ $\mu_{S0}$ is zero for all $T$. Qualitatively, these observations can be understood as follows: Since the baryon chemical potential couples to all quark flavors equally, cf. Eq. (2), increasing $\mu_B$ will also increase the number of strange quarks over antistrange quarks in the system. The strange chemical potential, on the other hand, favors antistrange over strange quarks and can therefore be tuned to compensate the strangeness generated by $\mu_B$. Obviously, if $\mu_B$ is zero, than $\mu_S$ also has to be zero to ensure strangeness neutrality. In the hadronic phase at small $\mu_B$ essentially all strangeness is carried by open strange mesons, in particular kaons and antikaons, since they can always be excited in the thermal medium. At small temperatures the Fermi surface of the baryons is very sharp while their Fermi energy is large, so at small $\mu_B$ and small $T$ essentially no baryons are excited. The thermally excited mesons will always have as much open strange as open antistrange in the case of isospin symmetry ($\mu_I = 0$) for $\mu_S = 0$. Hence, $\mu_{S0} \approx 0$ at small T and $\mu_B$. At large enough $\mu_B$ baryons can be excited and a finite $\mu_S$ becomes necessary to ensure strangeness neutrality. The corresponding strangeness will either be carried mostly by kaons (and $\kappa$) or by baryons, depending on $\mu_B$.

With increasing temperature the Fermi surface of baryons becomes increasingly diffused, facilitating the excitation of baryons. Hence, $\mu_S$ has to increase accordingly with temperature to maintain $n_S = 0$. This explains why $\mu_{S0}$ is monotonously increasing with temperature. In the vicinity of the phase transition, mesons and baryons start to dissolve into quarks. In the deconfined phase at large $T$ the quarks are only weakly interacting and hence flavor is decorrelated. In this case, there is an exact relation between baryon number and strangeness that directly follows from the coupling of $\mu_B$ and $\mu_S$ to the quarks in Eq. (6). This implies $\mu_{S0} = \mu_B/3$ in the deconfined phase. Since we find that the Polyakov loops are still smaller than one even at $T = 250$ MeV (characterizing the so called semi-QGP phase), complete deconfinement is not reached for highest temperatures in Fig. 4, which explains the the deviation of $\mu_{S0}$ from its asymptotic value.

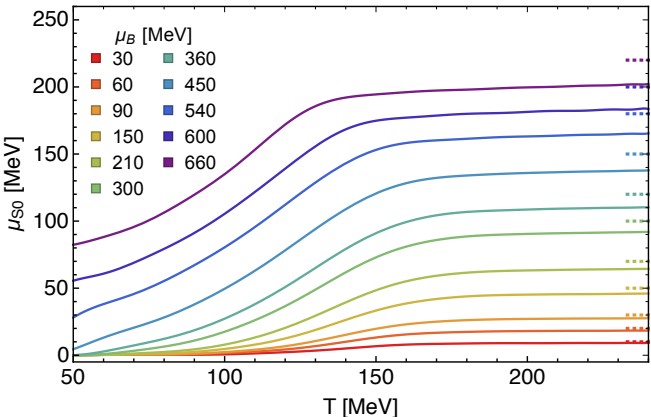

Figure 4: The strange chemical potential as a function of temperature at strangeness neutrality for various baryon chemical potentials. The asymptotic values for free quarks are indicated by the dotted lines at the right edge of the plot. $\mu_B$ is increasing from bottom to top.

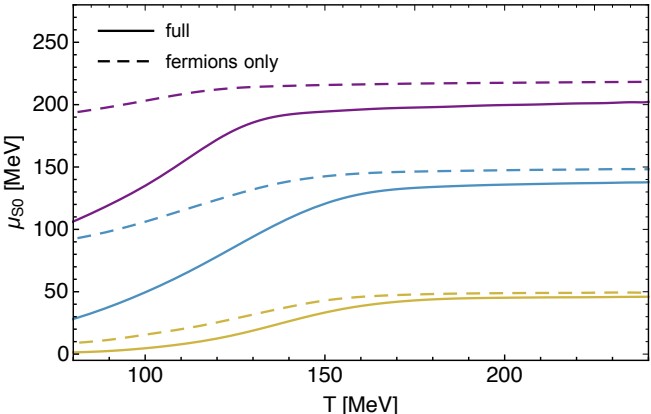

Figure 5: Comparison between our full result for $\mu_{S0}$ (solid lines) and Eq. (56) (dashed lines) for $\mu_B = 150$, 450 and 660 MeV (from bottom to top). The color coding for $\mu_B$ is the same as in Fig. 4.

Finally, we want to compare or findings to the predictions of a purely fermionic system. In [35] an intriguing relation between the Polyakov loops and the strangeness chemical potential at strangeness neutrality has been derived,

$$\mu_{S0}(T, \mu_B) \approx \frac{\mu_B}{3} - \frac{T}{2} \ln\left[\frac{\bar{L}(T, \mu_B)}{L(T, \mu_B)}\right].$$
(56)

The independence of the Polyakov loops on $\mu_S$ was assumed here. This equation can be derived from the quark contribution to the flow of the effective potential in Eq. (41). It provides a good measure for the effect of the quarks coupled to the gluon background field on strangeness neutrality. For the mean-field PNJL model studied in [35] it has be shown to be about 3% accurate. Potential deviations from this relation could be induced by a strong $\mu_S$-dependence of $\bar{L}/L$ and, most importantly, fluctuations of open strange mesons.

We show a comparison between our full result for $\mu_{S0}$ and Eq. (56) in Fig. 5. We have used the loops computed at $\mu_S = 0$ in this figure but have checked that the results depend only very mildly on this choice. While $L$ and $\bar{L}$ show a considerable dependence on $\mu_S$, their ratio does

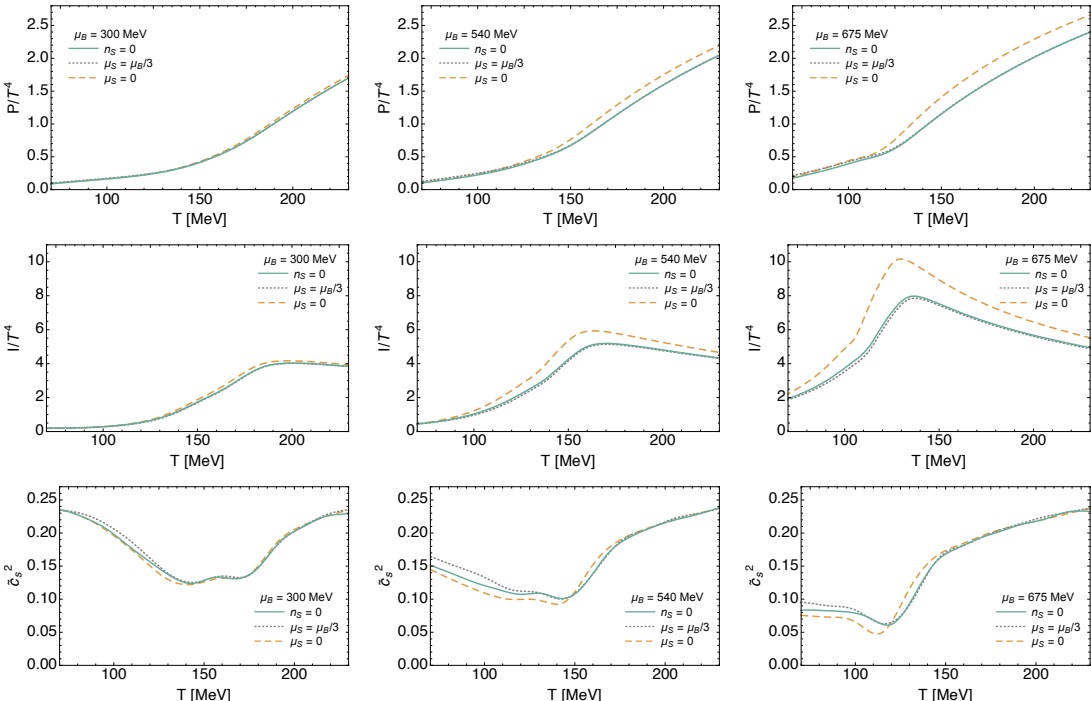

Figure 6: Comparison between the pressure (first row), the trace anomaly (second row) and the speed of sound squared (third row) at strangeness neutrality (solid blue line), at $\mu_S = 0$ (dashed orange line) and at $\mu_S = \mu_B/3$ (dotted gray line) for various $\mu_B$; see Eq. (54) for the definitions of these quantities.

not, excluding the former explanation for possible deviations. We see that Eq. (56) captures the qualitative trend of $\mu_{S0}$ quite well, but is quantitatively very inaccurate. At temperatures below the phase transition the difference can be attributed to the missing effect of open strange mesons in Eq. (56). This highlights the crucial importance of meson fluctuations for strangeness neutrality. At larger temperatures the asymptotic value $\mu_{S0} = \mu_B/3$ is rapidly reached with Eq. (56). The reason is that $\bar{L}/L \approx 1$ in this case, even though they are still smaller than one. As argued above, in our full result the asymptotic value is not reached since the system is in the semi-QGP phase. The heuristic relation does not capture this feature at all. We want to emphasize that $\bar{L}/L \to 1$ at large $T$ crucially depends on the parametrization of the Polyakov loop potential. In our case, Eq. (34), the Haar measure of the gauge group is implemented directly into the potential. This restricts the values of the loops to $L, \bar{L} \in [0, 1]$. For different parametrizations without the Haar measure the ordering $\bar{L} > L$ at finite $\mu_B$ persists for arbitrarily large temperatures, with loops larger than one. In this case, would also yield $\mu_{S0} < \mu_B/3$ at large $T$.

## 4.4 Strangeness neutrality and QCD thermodynamics

We can now use the results of the previous section to investigate the influence of the strangeness neutrality on thermodynamic quantities. To this end, we compare our results at $\mu_S = 0$ (dashed, orange) and $\mu_S = \mu_B/3$ (dotted, gray) to the ones at strangeness neutrality, $n_S = 0$ (solid, green), at various $\mu_B$. This is shown in Fig. 6. The first row shows the pressure, the second the trace anomaly and the third the speed of sound squared. For small baryon chemical potential, $\mu_B \lesssim 300$ MeV, the equation of state is not very sensitive to the chemical potentials since baryon excitations are highly suppressed. At small temperatures pion fluctuations dominate the equation of state in this case and hence the thermodynamic quantities are essentially

independent of $\mu_S$. At larger temperatures we find that the pressure and the trace anomaly are always smaller at strangeness neutrality than at $\mu_S = 0$. At larger $\mu_B$ this effect is more pronounced. The pressure and the trance anomaly start to grow at larger $T$ at strangeness neutrality as compared to $\mu_S = 0$, indicating that the QCD phase transition is shifted to larger temperatures. This is also apparent from the position of the minima of $c_s^2$, which approximately coincide with the pseudocritical deconfinement and chiral transition temperatures. Note that at $\mu_B = 675$ MeV we find $T_d \approx T_\chi$, so the two corresponding minima are degenerate. For $\mu_B = 675$ MeV the equation of state shows a sizable dependence on the strangeness. For the pressure we find a difference of about 20% between $\mu_S = 0$ and $n_S = 0$ at large temperatures and for the the trace anomaly even more than 35% in the transition region. The higher sensitivity of the trace anomaly is due to its direct dependence on the particle numbers. At strangeness neutrality, the baryon number is always smaller than at $\mu_S = 0$ for finite $\mu_B$ for all temperatures. This is as expected since finite $\mu_S$ leads to less strange particles in the system that can contribute to the baryon number.

In contrast to $p$ and $I$, the speed of sound squared shows the highest sensitivity in the small and intermediate temperature region. As discussed in Sec. 4.2, $p$ and $I$ are dominated by the increase in the number of degrees of freedom at the phase transition, while $c_s^2$ is not. In the hadronic regime we find a difference of about 30% between $\mu_S = 0$ and $n_S = 0$ at $\mu_B = 675$ MeV. This is also apparent from the comparison to the results at $\mu_S = \mu_B/3$. As argued in the previous section, $\mu_S = \mu_B/3$ enforces strangeness neutrality in case of uncorrelated quarks, i.e. deep in the deconfined phase. The results for $\mu_S = \mu_B/3$ and $n_S = 0$ should therefore become degenerate at large temperatures. This is also what we observe for the thermodynamic quantities. Since $\mu_{S0}$ is already close to its asymptotic value at $T_\chi$, cf. Fig. 4, they are already very similar close to the chiral transition for $\mu_S = \mu_B/3$ and $n_S = 0$. The pressure and the trace anomaly show only very small differences between $\mu_S = \mu_B/3$ and $n_S = 0$ at small temperatures. $c_s^2$ shows a stronger sensitivity to the strangeness below the chiral phase transition. $\mu_S = \mu_B/3$ results in a larger and $\mu_S = 0$ in a smaller speed of sound in the hadronic phase as compared to the result at strangeness neutrality. This ordering is inverted for the pressure and the trace anomaly.

Overall, we found that the equation of state becomes increasingly sensitive to strangeness with increasing baryon chemical potential. At $\mu_B = 675$ MeV, where the transition is still a crossover in our model, the effects of strangeness neutrality as compared to vanishing strange chemical potential become as large as about 30%.

## 4.5   Strangeness neutrality and the phase structure

As already indicated by the results in the previous section, strangeness has a sizable effect on the phase structure at finite baryon chemical potential. In the left plot of Fig. 7 we show the phase diagram of the chiral transition as defined by the inflection point of the subtracted chiral condensate, Eq. (53), at strangeness neutrality (solid line) and at vanishing strangeness chemical potential (dashed line). We see, as already concluded in the previous section, that strangeness neutrality leads to a larger critical temperature as compared to $\mu_S = 0$. The effect increases with increasing $\mu_B$, but leads to only about 6% difference in $T_\chi$ at the largest baryon chemical potential and is therefore very small. However, since the transition is a crossover for the parameters considered here, it is more sensible to compare the global structure of the order parameters. To this end, we computed the relative difference between the subtracted condensate at strangeness neutrality and at vanishing strange chemical potential,

$$\frac{\Delta_{LS}\big|_{n_S=0} - \Delta_{LS}\big|_{\mu_S=0}}{\Delta_{LS}\big|_{n_S=0}} . \tag{57}$$

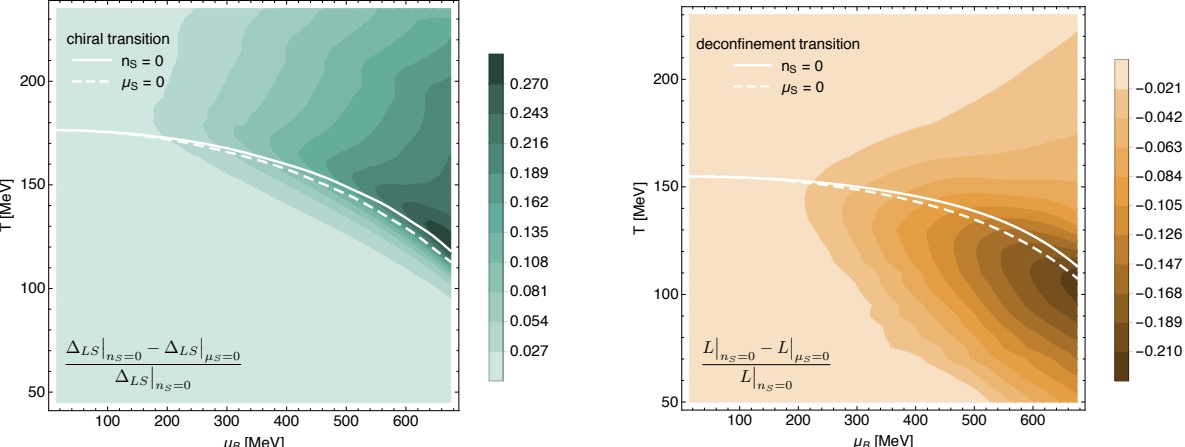

Figure 7: Left: Relative error of the subtracted condensate for strangeness neutrality and $\mu_S = 0$. The solid and dashed lines indicate the chiral phase boundary as defined by the inflection point of $\Delta_{LS}(T)$ for $n_S = 0$ and $\mu_S = 0$ respectively. Right: The same for the Polyakov loop. Here, the solid and dashed lines indicate the deconfinement phase boundary as defined by the inflection point of $L(T)$ for $n_S = 0$ and $\mu_S = 0$.

The result is given by the density profile in the left plot of Fig. 7. The darker the color, the larger the difference. It shows where the chiral phase structure is most sensitive to strangeness. Similar to our findings for the pressure, the subtracted chiral condensate is most sensitive at intermediate to large $\mu_B$ and above the critical temperature. In the hadronic phase, strangeness neutrality does not have a big effect on the chiral order parameter. Even though the effect of strangeness neutrality on the inflection point of the order parameter is rather small, we find deviations of up to about 27% in the difference defined in Eq. (57). The relation $\Delta_{LS}\big|_{n_S=0} \geq \Delta_{LS}\big|_{\mu_S=0}$ holds for all $T$ and $\mu_B$ considered here. Baryon effects (relative to meson effects), which tend to make the crossover steeper, are partly suppressed at strangeness neutrality since $\mu_S > 0$ effectively reduces strange baryon contributions. The chiral condensate therefore melts slower at strangeness neutrality.

A similar conclusion can be drawn for the deconfinement transition. In the right plot of Fig. 7 we show the deconfinement transition as defined by the inflection point of the Polyakov loop, Eq. (24), at strangeness neutrality (solid line) and at vanishing strange chemical potential (dashed line). The antiloop $\bar{L}$ gives essentially the same critical temperature. As for the chiral transition, the pseudocritical temperature becomes slightly larger at $n_S = 0$ as compared to $\mu_S = 0$, where the difference increases with increasing $\mu_B$. We also computed the relative difference

$$\frac{L\big|_{n_S=0} - L\big|_{\mu_S=0}}{L\big|_{n_S=0}}, \tag{58}$$

and the result is given by the density profile in the right plot of Fig. 7. Again, we find that the deviation grows with $\mu_B$ but this time is largest in the hadronic regime right below the phase boundary. Recalling that the deconfined phase corresponds to chiral symmetry *restoration* and center symmetry *breaking*, we conclude that both for the chiral and the deconfinement order parameter, the transition region at large $\mu_B$ towards the respective symmetry *restored* phase is most sensitive to strangeness. For the Polyakov loops we always find $L\big|_{n_S=0} \leq L\big|_{\mu_S=0}$. The overall effect on the deconfinement transition is a bit smaller than on the chiral transition, but still about 20%. These findings might suggest that the results for the effect of strangeness neutrality on the thermodynamic quantities in the previous section could be attributed to the

pressure and the trance anomaly being more sensitive to the chiral transition, while the speed of sound is more sensitive to the deconfinement transition.

Finally, we studied how strangeness neutrality affects the isentropes in the phase diagram. They are defined by trajectories of constant $s/n_B$. Without dissipation, i.e. the ideal case, the hydrodynamic evolution of the quark-gluon plasma is along such isentropes. This is due to the fact that without dissipation and only strong interactions, both the entropy density and the baryon number are conserved in the hydro evolution. Even though it is established by now that the QGP is not an inviscid fluid, given the small shear viscosity over entropy density of the QGP suggest by hydrodynamic simulations of heavy-ion collisions, the isentropes still provide a good estimate for the approximate path that the QGP in its late stages takes through the phase diagram.

Our results are shown in Fig. 8. The orange dashed line corresponds to $\mu_S = 0$ and the solid blue line shows the isentropes at strangeness neutralities for various fixed ratios $s/n_B$. The isentropes show a very characteristic behavior: they have positive slope in the phase diagram above the phase transition and a negative slope below. In the transition region, the slope changes sign, with a slower 'turning' of the isentropes at smaller $\mu_B$, where the crossover region is wider. We find this kink even at large $s/n_B$. Interestingly, in studies of the isentropes within two-flavor QM and PQM models such a kink only occurs for small $s/n_B$ [40,96]. Hence, the sensitivity of the isentropes to the phase transition at large $s/n_B$ can be attributed to srangeness.

The behavior of the isentropes in the hadronic phase is dictated by the Silver-Blaze property of QCD. At $T = 0$ and $\mu_B \lesssim 3m_l$ the baryon number has to vanish. Hence, the isentropic curves bend toward larger $\mu_B$ with decreasing $T$. The difference between $n_S = 0$ and $\mu_S = 0$ is small at small temperatures because the lightest baryonic resonance does not carry strangeness. Since the system is in the semi-QGP phase above the phase transition, the entropy density has not reached its asymptotic value yet and is hence still growing with $T$. The baryon number, on the other hand, has a maximum at the chiral phase transition and slowly decreases with increasing temperature above $T_\chi$. Hence, the isentropes bend towards larger $\mu_B$ with increasing $T$ above the phase transition. The regions where the isentropes turn therefore clearly indicate the transition region. Since the baryon number at strangeness neutrality is systematically smaller than for $\mu_S = 0$ at a given $\mu_B$, the bending of the isentropes above the phase transition is stronger at strangeness neutrality. We also find that the isentropes at strangeness neutrality are systematically shifted to the right. Qualitatively, this can be understood from the fact that the baryon number decreases with increasing $\mu_S$. This effect dominates over the corresponding effect on the entropy density (which behaves very similar to the pressure in Fig. 6). Thus, larger $\mu_B$ is necessary to ensure a fixed $s/n_B$ at strangeness neutrality.

# 5 Summary

Strangeness neutrality is a crucial property of the matter created in heavy-ion collisions. We studied its impact on QCD thermodynamics and the phase structure. To this end, we set up a 2+1 Polyakov loop enhanced quark-meson model that captures the dynamics of mesons, quarks and, to some extent, baryons in a gluon background field at finite baryon and strangeness chemical potential. We demonstrated by comparing to available lattice data that this works very well for the QCD equation of state not only at vanishing chemical potential, but also at finite $\mu_B/T$.

Demanding that the strangeness number is always zero implicitly defines a corresponding strange chemical potential as a function of temperature and baryon chemical potential. We computed resulting function $\mu_{S0}(T, \mu_B)$. Its non-trivial functional form has a transparent interpretation in terms of competing strange meson and baryon dynamics at finite baryon

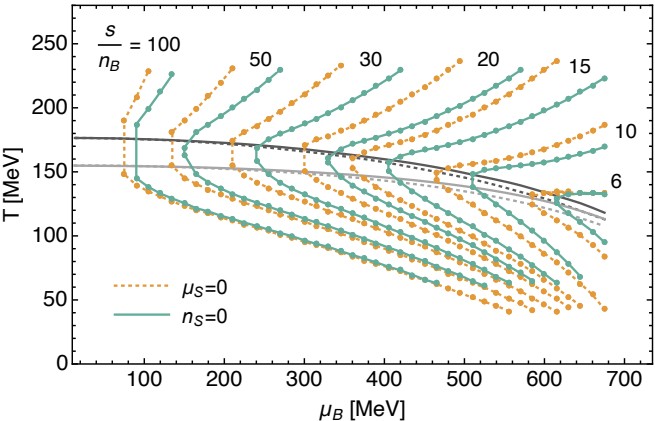

Figure 8: Isentropes in the phase diagram. The dark and light gray lines are the chiral and deconfinement phase boundaries respectively.

chemical potential and is therefore intimately tied to confinement. We compared these results to the purely fermionic case, i.e. where only quark and baryon dynamics are taken into account, and found huge discrepancies. This highlights the crucial importance of open strange meson dynamics for the accurate description of strangeness physics and the freeze-out conditions of heavy-ion collisions.

We used our results for $\mu_{S0}(T, \mu_B)$ to compute QCD thermodynamics and the phase structure at strangeness neutrality. The effect of the strangeness content of the QCD medium on its thermodynamics is certainly interesting on its own right, but also very important as an input for, e.g., the hydrodynamic description of heavy-ion collisions. The comparison of our results at vanishing density to lattice QCD results show very good agreement, even for the highly non-trivial speed of sound. To assess the effect of strangeness neutrality we confronted results on the equation of state at fixed strange chemical potential, where we have chosen $\mu_S = 0$ and $\mu_B/3$, to the equation of state at strangeness neutrality. For reasons related to the range of validity of our model (see App. B) we restricted our analysis to $\mu_B \in \{0, \dots, 675\}$ MeV but note that this covers the region probed by current beam energy scan experiments [15] (assuming that the translation of the beam energy to the baryon chemical potential based on the hadron resonance gas is correct). Our results show that the relevance of strangeness neutrality grows with increasing baryon chemical potential and the difference between strangeness neutrality and $\mu_S = 0$ can be as large as about 30% at $\mu_B = 675$ MeV, in particular for the trace anomaly and the speed of sound squared.

We find a similar sensitivity of the chiral and deconfinement phase transitions on strangeness. Overall, the pseudocritical temperatures of both transition are larger at strangeness neutrality than at vanish strange chemical potential. Hence, strangeness neutrality 'delays' the transition to the QGP. Again, while the effect is small at small $\mu_B$ and becomes considerable at larger $\mu_B$. This can be attributed to a suppression of symmetry-breaking fermionic fluctuations in the strange sector due to finite $\mu_S$. Due to their distinct sensitivity to the phases of QCD and the related thermodynamics, the isentropes, which provide a good estimate for the path of the hydrodynamic evolution of the QGP though the phase diagram, also turned out to be affected by strangeness neutrality significantly.

In summary, we have demonstrated that the QCD equation of state and its phase structure are highly sensitive to the strangeness content of the medium. For the accurate description of heavy-ion collisions at varying beam energies it is indispensable to take this into account. The underlying physics is very intriguing since the strangeness neutrality condition $n_S = 0$ is sensitive to various characteristic properties of QCD, namely the interplay of meson and

baryon dynamics at finite chemical potential as well as the chiral and deconfinement phase structure. The present results facilitate the computation of fluctuation observables in heavy-ion collisions, such as higher cumulants of baryon number and strangeness distributions including off-diagonal cumulants, under more realistic conditions.

Towards a more realistic equation of state, the next crucial step is to also account for the freeze-out condition related to the initial charge of the colliding nuclei by taking finite isospin chemical potential into consideration. Also in this case, beyond mean-field effects and in particular pion fluctuations will certainly be very important. Concerning the model, the most relevant improvements are the inclusion of effects beyond LPA which have a high impact on quark and meson dynamics, and the incorporation of dynamics in the gauge sector which allow for a self-consistent computation of the Polyakov loop potential. The latter point might remedy the thermodynamic inconsistency of the PQM model at large $\mu_B$ discussed in the appendix and thus allow for an extension of the present work towards the critical endpoint of QCD. Then, (off-diagonal) cumulants of baryon number and strangeness distributions will also become accessible.

## Acknowledgments

We thank Mario Mitter, Robert D. Pisarski, Vladimir Skokov and Patrick Steinbrecher for discussions as well as Jens Braun and Bernd-Jochen Schaefer for valuable comments on our manuscript. We also thank the members of the fQCD collaboration [1] for discussions and work on related projects. We thank Dirk H. Rischke for helpful explanations regarding the speed of sound. F.R. is supported by the Deutsche Forschungsgemeinschaft (DFG) through grant RE 4174/1-1. W.F. is supported by the National Natural Science Foundation of China under Contracts Nos. 11775041. This work is supported by the ExtreMe Matter Institute (EMMI) and the grant BMBF 05P12VHCTG. It is part of and supported by the DFG Collaborative Research Centre "SFB 1225 (ISOQUANT)".

## A   Details on the Polyakov loop potential

Here we provide the details on the Polyakov loop potential $U_{\text{glue}}$. For $x = (a, c, d)$ the temperature dependent coefficients in Eq. (34) are of the form

$$x(T) = \frac{x_1 + x_2/t + x_3/t^2}{1 + x_4/t + x_5/t^2},$$

$$b(T) = b_1 \, t^{-b_4} \big(1 - e^{b_2/t^{b_3}}\big),$$

(59)

where the parameters have been determined in [77] and are shown in Tab. 2. $t = t_{\text{red}} + 1$ with $t_{\text{red}} = \alpha_t(T - T_0)/T_0$. $T_0$ is the deconfinement temperature of the pure gauge theory, while $\alpha_t$ is a parameter that controls the speed of the transition. Due to unquenching effects, both parameters deviate from the values of the pure gauge theory, $T_{0,\text{YM}} = 276$ MeV and $\alpha_{t,\text{YM}} = 1$. Since the QCD transition has a smaller critical temperature and a smoother transition, one generally expects $T_0 < T_{0,\text{YM}}$ and $\alpha_t < \alpha_{t,\text{YM}}$. In [97] $\alpha_t = 0.57$ has been determined. However, since this depends on the number of flavors, the truncation and the parametrization of the Polyakov loop potential, we will consider both $\alpha_t$ and $T_0$ as free parameters here. They can be

---

[1] J. Braun, L. Corell, A. K. Cyrol, W.-j. Fu, C. Huang, M. Leonhardt, M. Mitter, J. M. Pawlowski, M. Pospiech, F. Rennecke, C. Schneider, R. Wen, N. Wink, S. Yin.

Table 2: Fit parameters of the Poyakov loop potential defined in Eqs. (34) and (59). These are taken from [77].

| $a_1$ | $a_2$ | $a_3$ | $a_4$ | $a_5$ |
|---|---|---|---|---|
| -44.14 | 151.4 | -90.0677 | 2.77173 | 3.56403 |
| $b_1$ | $b_2$ | $b_3$ | $b_4$ | |
| -0.32665 | -82.9823 | 3.0 | 5.85559 | |
| $c_1$ | $c_2$ | $c_3$ | $c_4$ | $c_5$ |
| -50.7961 | 114.038 | -89.4596 | 3.08718 | 6.72812 |
| $d_1$ | $d_2$ | $d_3$ | $d_4$ | $d_5$ |
| 27.0885 | -56.0859 | 71.2225 | 2.9715 | 6.61433 |

determined, e.g., by fitting the pressure to the lattice result at vanishing density. The other fit parameters of the potential are given by their YM values and are given in Tab. 2:

The inclusion of finite chemical potentials to the gauge sector can be achieved along the lines of [29,41]. It is constructed phenomenologically from the identification of $\Lambda_{QCD}$ in the one-loop beta function of QCD at large density (HTL/HDL) with the flavor dependent modification of the critical temperature. This suggests the following modification of $T_0$, [29],

$$T_0(N_f, \mu) = T_\tau e^{-1/(\alpha_0 b_\mu)}, \tag{60}$$

where $T_\tau = 1.77$ GeV sets the renormalization scale with the corresponding coupling $\alpha_0 = 0.304$ for $N_f = 0$. $b_\mu$ encodes the flavor and chemical potential dependence of beta function:

$$b_\mu = b_0 + \frac{16}{\pi}\left[2\frac{\mu^2}{(\hat{\gamma} T_\tau)^2}\Delta n_l + \frac{(\mu - \mu_S)^2}{(\hat{\gamma} T_\tau)^2}\Delta n_s\right]. \tag{61}$$

$b_0$ can be chosen either to be the well-known one-loop QCD beta function coefficient, $b_0 = (11N_c - 2N_f)/(6\pi)$, or, in the spirit of $T_0$ as an approximation dependent free parameter, to also be a free parameter. The second term in Eq. (61) is constructed such that the chiral and deconfinement transition agree at finite $\mu$ at mean-field in the two flavor PQM [29]. $\hat{\gamma}$ can be used as an additional parameter to control the curvature of the deconfinement phase transition. We use $\hat{\gamma} = 1$ for the time being. The distributions $\Delta n_{l/s}$ are introduced in order to maintain the Silver Blaze property at vanishing temperature. For $\Delta n_{l/s} = 1$ the above parametrization would yield a $\mu$-dependent equation of state at vanishing temperature. Under the requirement that $\Delta n_{l/s} \to \Theta(\mu - M_{l/s})$ at vanishing temperature, we define

$$\Delta n_l = \frac{1}{e^{3(M_l - \mu)/T} + 1} + \frac{1}{e^{3(M_l + \mu)/T} + 1}\frac{2}{e^{3M_l/T} + 1},$$

$$\Delta n_s = \frac{1}{e^{3(M_s - \mu + \mu_S)/T} + 1} + \frac{1}{e^{3(M_s + \mu - \mu_S)/T} + 1} - \frac{2}{e^{3M_s/T} + 1}, \tag{62}$$

where $M_{l/s}$ are renormalized vacuum masses of the light and strange quarks.

## B  Thermodynamics at large $\mu$

Throughout this work, we have used $\mu_B \leq 675$ MeV. This is below the critical endpoint of the model, which would certainly be interesting to study also in the context of this work. We find

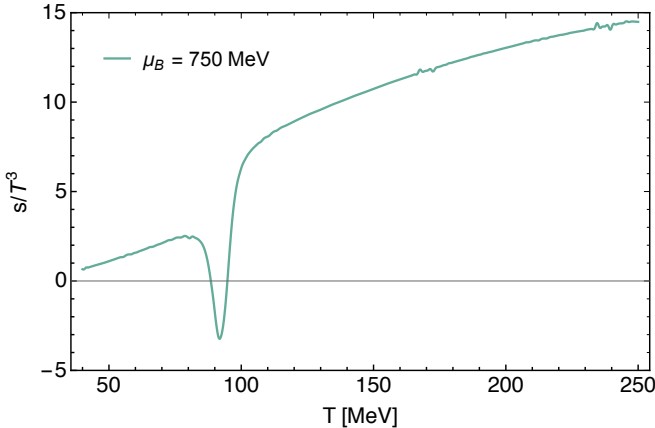

Figure 9: Entropy density at $\mu_B = 750$ MeV and $\mu_S = 0$.

that starting at $\mu_B \gtrsim 700$ MeV the pressure develops an increasingly strong non-monotonoticity with increasing $\mu_B$ in the vicinity of the phase transition. This eventually leads to a negative entropy density in this region, as shown in Fig. 9 at $\mu_B = 750$ MeV and $\mu_S = 0$. We explicitly checked that this is independent of the parametrization of the loop potential. The origin of this behavior can be traced back to the contribution of the gauge sector to the pressure,

$$p\big|_{\mathrm{glue}} = -U_{\mathrm{glue}}(L, \bar{L}), \tag{63}$$

where the Polyakov loops are part of the solutions of the equations of motion. We show this contribution at $\mu_B = 0$ and $\mu_B = 750$ MeV for $\mu_S = 0$ in Fig. 10. This contribution is negative and has a minimum at around the chiral transition temperature. We see that the larger $\mu_B$ the larger this negative contribution tho the pressure becomes. For the baryon chemical potentials used in the main part of this work, where the pressure is always monotonously increasing, this negative contributions can be interpreted as the suppression of hadronic contributions to the pressure in the transition region due to deconfinement. This effect then is clearly overestimated at large $\mu_B$, leading to unphysical thermodynamics.

This problem originates in a combination of potential effects:

Firstly and most prominently, the construction of the Polyakov loop potential we use in this work, Eq. (34), is based on the pressure, the expectation values of the Polyakov loops and their two-point correlators [77]. This corresponds to Taylor expansion to second order of the potential about the minimum. The pressure is the value of the potential, the Polyakov loop expectation value determines the location of the minimum and the two-point correlator determines the curvature in the minimum. Further information on the global form of the potential comes from the temperature dependence of the parameters and the Haar measure of the loop. Evidently, this does not fully constrain the potential away from the Yang-Mills minimum. Moreover, the potential is best constrained for $L = \bar{L}$. The further away from the expansion point the potential has to evaluated, in particular for $L \neq \bar{L}$, the less constrained it is. This could be cured by either taking into account higher correlation functions of the loops in an extension of [77], or by using a self-consistent $\bar{A}_0$-potential from the FRG [69, 72, 98].

Secondly, the effect of matter fluctuations is only taken into account effectively by a simple quark flavor and chemical potential dependent rescaling of the potential as discussed in App. A. While this works well at small chemical potential, it might be too simple at large chemical potential. This problem could be cured by a self-consistent FRG computation as mentioned above.

Thirdly, for large chemical potentials and temperatures the initial conditions depend on these external parameters. Within the present approximation this is discussed in Sec. 4.1. More

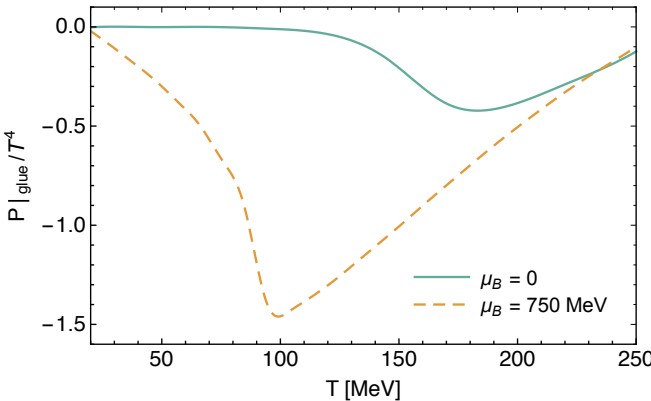

Figure 10: Contribution of the Polyakov loop potential to the pressure.

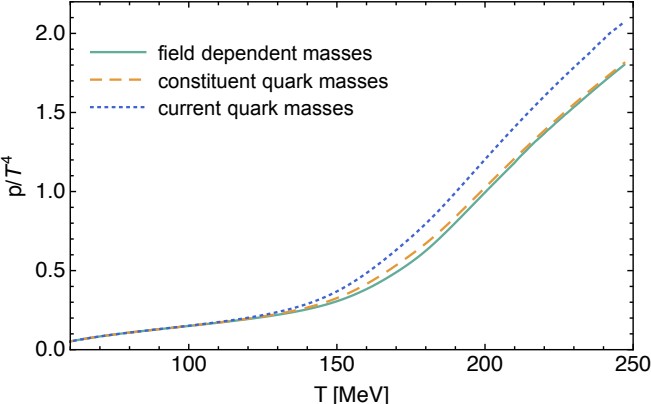

Figure 11: The pressure at $\mu = 0$. The solid green line is computed with the full field dependence of $\Delta\Gamma_\Lambda$. The dotted blue line shows the results with $m_l = 3.6$ MeV and $m_s = 95$ MeV. The dashed orange line correspond to $m_l = 300$ MeV and $m_s = 430$ MeV.

generally, information from QCD at large energy scales are required, see e.g. [99].

Lastly, for large chemical potentials it might be possible that the free energy is minimized by an inhomogeneous solution. Consequently, our solution on a homogenous background could potentially lead to a negative contribution to the pressure, see, e.g., [100–103] for studies within (P)NJL and QM models. Given the explicit analysis done below and due to the occurrence of this problem already at moderate chemical potential, this is unlikely to be the origin of the problem in the present case. Furthermore, by using a Fierz-complete basis for the four-quark interaction channels within a NJL model, it has been shown in [104, 105] that other channels, for instance isoscalar-vector and diquark channels, become relevant for the phase structure at finite baryon chemical potential. Since we only account for the scalar-pseudoscalar channel in this work (cf. Sec. 2.2), we might miss some relevant effects at larger chemical potential.

The problems discussed above manifest themselves in the gluon contribution to the pressure in the present work. In Fig. 10 we show the contribution of $U_{\text{glue}}$ to the pressure at $\mu_B = 0$ and $\mu_B = 750$ MeV at vanishing strangeness chemical potential. Since the deconfinement transition in $SU(3)$ Yang-Mills theory is of first order, $U_{\text{glue}}$ is normalized such that its minimum is at zero for $T < T_0$. The Polyakov loops are also exactly zero in this case, $L^{\text{YM}} = \bar{L}^{\text{YM}} = 0$. In the present work, and Polyakov-loop enhanced models of QCD in general, the deconfinement

transition is a crossover and the Polyakov loops $L$, $\bar{L}$ are always non-zero. This means that $U_{\text{glue}}$ is probed away from its normalized minimum, so while $U_{\text{glue}}(L^{\text{YM}}, \bar{L}^{\text{YM}}) = 0$ for $T < T_0$ one has $U_{\text{glue}}(L, \bar{L}) > 0$. For increasing $\mu_B$ the Polyakov loops in QCD become larger and also unequal. So, as discussed above, we probe the potential in a region that is not well described by the present parametrization. This explains our observation in Fig. 9 and why we refrain from doing computations at too large $\mu_B$.

## C  Field dependence of the initial conditions

Here, we check the effect of the meson field dependence of $\Delta\Gamma_\Lambda$ as discussed in Sec. 4.1. One may argue that it is sufficient to use the current quark, or even vanishing, masses in Eq. (47) instead of resorting to background field dependent quark masses. However, it turns out that this is quantitatively very inaccurate in the present case. The medium-dependent corrections for the effective potential become relevant well before the quark masses reach their current values. In particular in the LPA, the quarks approach their current mass very slowly above $T_c$, if at all, so that they are reached well above the temperatures relevant here. As a result, setting $m_l \approx 3.6$ MeV and $m_s \approx 95$ MeV leads to a significant overestimation of the in-medium corrections to the initial action. The same is true for the case where the initial quark masses that follow from the initial parameters in Tab. 1, which are about a factor of 2-3 larger than the PDG current masses, are used.

However, it turns out that using the vacuum constituent quark masses, $m_l \approx 300$ MeV and $m_s \approx 430$ MeV works quite well. This is shown in Fig. 11. As argued in Sec. 4.1, the most accurate determination of the equation of state is by using the background field dependent quark masses in the in-medium corrections of the initial conditions. This is the solid green line in the figure. Using the current quark masses leads to a considerable overestimation of the pressure, as shown by the dotted blue line. The dashed orange line shows the result with the constituent quark masses and we see that it gives a very accurate result. The error of this procedure is largest in the transition region, where it is about 8%. We have checked explicitly that these findings are also true at finite chemical potentials. The advantage of the field independence is obviously that $\Delta\Gamma_\Lambda$ only enters in the initial pressure. Only $\omega_{00,\Lambda}$ in Eq. (51) receives a correction from $\Delta\Gamma_\Lambda$. The numerical integration of higher derivatives of Eq. (47) for the correction to the higher Taylor coefficients becomes unnecessary and irrelevant operators can be se to zero at the initial scale. At order $\phi^{10}$ this results in a speed-up by a factor of two to three with the numerical integration we implemented. Hence, given this large numerical speedup we accept the relatively small systematic error in our results on thermodynamics.

We would like to emphasize that these results apply to the fixed background Taylor expansion we used to solve the flow equation of the effective potential [42,49] and might not be directly transferrable to other methods. This is due to the fact that we expand the effective potential about its temperature and chemical potential dependent IR minimum. Using the current quark masses for the in-medium corrections of the initial effective potential is therefore consistent with our expansion scheme. For a more general discussion on this matter we refer to [90].

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
