# Peer review of "Strangeness Neutrality and QCD Thermodynamics"

_SciPost Physics, doi:SciPost Phys. Core 2, 002 (2020)_

## Round 1 · Referee Report · Anonymous · 2018-11-6

Strengths

1 - A three flavour Polyakov-quark-meson model analysis within a FRG framework in local potential approximation including baryon and strangeness chemical potential

2 - Illustrative examples Figs 3 - 5 for the strangeness density as well as strange chemical potential as functions of temperature and chemical potentials.

3 - Discussion of the influence of strangeness neutrality on the isentropes in the phase diagram (Fig 8).

Weaknesses

1 - lengthy explanation of three flavour QCD at low energy, in particular parts of Sections II and III which are ultimately not used in the work.

Report

The manuscript discusses the impact of strangeness neutrality on the QCD thermodynamics and its phase structure. The dynamics of the quarks and mesons and to some extent also the baryons at finite baryon and strangeness chemical potentials are tackled by a renormalization group approach with a 2+1 quark flavour quark-meson model truncation augmented with a Polyakov loop. The authors conclude that the impact of strangeness neutrality on the thermodynamics and on the equation of state is sizeable.

The topic of this work is contemporary and very interesting. The reference list is exhaustive and gives a fair account of the existing literature on these issues. However, before I can recommend the manuscript for publication in SciPost the following points need to be addressed by the authors:

1) In Eq(16) it is stated that this expression at some finite RG scale k is the total thermodynamic potential which seems to be a little misleading. For the physical quantities the solution of a gap equation is needed which is not mentioned.

2) The setup described in Sec II und III, in particular the one in Sec III, are quite lengthy, to some extent irrelevant for the present work and off-putting to the reader. Let me give a concrete example: on page 7 in Sec B wave function renormalizations are introduced but later these quantities are not even used since a local potential approximation (LPA) is employed. This is distracting from what is actually done and should be adjusted accordingly. Furthermore, it is not clear whether the chosen flat Litim regulator is a valid regulator beyond a local potential approximation. This is not addressed, and in the absence of a solution it suggests that general discussions beyond LPA should be refrained from.

3) In Fig 7 the chiral transition defined by the inflection point of the subtracted chiral condensate at strangeness neutrality and at vanishing strangeness chemical potential is shown.
From this result it is stated that strangeness neutrality leads to a larger critical temperature as compared to $\mu_s =0$. This statement is not clear since both curves are almost identical.

4) In Fig 8 it would be helpful to include the crossover line.

5) In App B it is stated that the deconfinement transition in SU(3) Yang-Mills theory is of second-order in contradiction to other findings.

6) Fig 4 the colour coding is hard to read and several word duplications like the the etc appear in the main body.

Requested changes

1) In Eq(16) it is stated that this expression at some finite RG scale k is the total thermodynamic potential which seems to be a little misleading. For the physical quantities the solution of a gap equation is needed which is not mentioned.

2) The setup described in Sec II und III, in particular the one in Sec III, are quite lengthy, to some extent irrelevant for the present work and off-putting to the reader. Let me give a concrete example: on page 7 in Sec B wave function renormalizations are introduced but later these quantities are not even used since a local potential approximation (LPA) is employed. This is distracting from what is actually done and should be adjusted accordingly. Furthermore, it is not clear whether the chosen flat Litim regulator is a valid regulator beyond a local potential approximation. This is not addressed, and in the absence of a solution it suggests that general discussions beyond LPA should be refrained from.

3) In Fig 7 the chiral transition defined by the inflection point of the subtracted chiral condensate at strangeness neutrality and at vanishing strangeness chemical potential is shown.
From this result it is stated that strangeness neutrality leads to a larger critical temperature as compared to $\mu_s =0$. This statement is not clear since both curves are almost identical.

4) In Fig 8 it would be helpful to include the crossover line.

5) In App B it is stated that the deconfinement transition in SU(3) Yang-Mills theory is of second-order in contradiction to other findings.

6) Fig 4 the colour coding is hard to read and several word duplications like the the etc appear in the main body.

---

## Round 1 · Referee Report · Anonymous · 2018-11-12

Strengths

First investigation of enforcing strangeness conservation in the context of functional renormalization group applied to PQM.

Weaknesses

No error bars are presented on the final results which indicate the level of theoretical uncertainty.

Report

Overall I find the paper to be well-written, however, I have some questions that need to be addressed by the authors.

Requested changes

(1) Since the authors work in a FRG-improved PQM there are only mesonic and quark degrees of freedom present. This leads me to two points of confusion: (a) What happens at very large temperatures to the pressure and trace anomaly? Does the result for the pressure go to the Stefan-Boltzmann limit for QCD as T -> infty? It would seem to me that the answer is no because there are no dynamical gluons (extra Nc^2 -1 massless degrees of freedom). If I can't go to T-> infty, what is the highest temperature at which I would expect such a model to be trustworthy? (b) What is the impact of throwing out baryons/resonances as one approaches Tc? Traditionally one expects to be sensitive to higher mass states as one approaches Tc.

(2) There are no uncertainties lies in Tables I or II. These were determined by some fitting procedure. It would be nice to know the uncertainties and the impact these uncertainties have on the final results.

(3) In the intro the authors state that they will "go beyond mean field level". This seems to be true for mesonic fluctuations but not for the Polyakov loop itself. This should be stated clearly in the intro elsewhere where it's stated that you are going beyond mean field level.

---

## Round 2 · Referee Report · Anonymous (Referee 1) · 2020-1-16

Report

The authors of the present manuscript have addressed almost all of my concerns in a satisfactory manner.

However, though the authors agree with my second point concerning the lengthy setup in Sec II and III, they have only made marginal changes. This is regrettable since the lengthy discussions obscure the core of this work. I do not think it is always a matter of taste as to whether one should make effort to improve accessibility, readability and, subsequently, impact.

Nevertheless, to avoid any further delay this work should be published now.

---

## Round 2 · Author Response

We thank the referees for their helpful comments and suggestions. We apologize for the belated reply and want to assure that this delay is in no way an indication of a lack of confidence in our results, but it may be attributed to negligence.

In the following, we quote the points raised by the referees and comment on them.

Referee 1

(1) "In Eq. (16) it is stated that this expression at some finite RG scale k is the total thermodynamic potential which seems to be a little misleading. For the physical quantities the solution of a gap equation is needed which is not mentioned."

The referee is right. To clarify this, we added/modified the corresponding equations on p. 4 (Eqs. (16) and (17)). In addition, we added that the (symmetric part) of the meson effective potential explicitly depends on the gluon background field \bar{A} in Eqs. (13) and (14).

(2) "The setup described in Sec II und III, in particular the one in Sec III, are quite lengthy, to some extent irrelevant for the present work and off-putting to the reader. Let me give a concrete example: on page 7 in Sec B wave function renormalizations are introduced but later these quantities are not even used since a local potential approximation (LPA) is employed. This is distracting from what is actually done and should be adjusted accordingly. Furthermore, it is not clear whether the chosen flat Litim regulator is a valid regulator beyond a local potential approximation. This is not addressed, and in the absence of a solution it suggests that general discussions beyond LPA should be refrained from."

We agree that the discussion of our setup is lengthy in some parts. Whether or not this is off-putting is certainly a matter of taste. The physical motivation of considering strangeness neutrality lies in the specific conditions found in heavy-ion collisions and is therefore phenomenological in nature. We attempted to make some parts of our work more accessible in order to potentially appeal to a wider audience. To this end, we provide detailed explanations regarding the inclusion of open strange mesons and how baryon effects are captured effectively.

Furthermore, whether or not general discussions within the paper should also be strictly within the approximations of the numerical implementation is also a matter of taste. Without any question, it should become clear which approximations are used for the final results. We believe that our presentation is not misleading in this respect. To be specific, we decided to be slightly more general in the discussion of chiral invariants in the presence of a strangeness chemical potential at the end of Sec. II.B. To us, this discussion is useful since it concerns the validity of our construction of the effective potential. The other point is the discussion of the open strange meson contribution to the flow of the effective potential, where we outlined the derivation in case the meson wave function renormalization is running. Since we do not discuss the effects of running wave function renormalizations any further, we agree with the referee that the more general discussion is unnecessary at this point, so we removed it from the discussion in and around Eqs. (38) and (40).

Nonetheless, we would like to comment on the referee's statement regarding the validity of the Litim regulator. It is most certainly a valid regulator even beyond LPA in that it fulfills all criteria for a valid regulator and does not lead to any pathologies in the RG flow. Beyond that, one may ask whether or not it is the optimal choice for a regulator, in the sense that it minimizes truncation errors. This important question is not subject of the present work.

(3) "In Fig 7 the chiral transition defined by the inflection point of the subtracted chiral condensate at strangeness neutrality and at vanishing strangeness chemical potential is shown. From this result it is stated that strangeness neutrality leads to a larger critical temperature as compared to μS = 0. This statement is not clear since both curves are almost identical."

The effect on the pseudocritical temperature itself is indeed very small. However, as we also show in Fig. 7, the effect on the order parameters as functions of T and \mu_B is sizable at larger \mu_B. Hence, strangeness neutrality is relevant for the melting of the order parameters in the crossover region. In this region, the global form of the order parameters is a more sensible measure for the restoration of symmetry than the pseudocritical temperatures. To clarify this point, we emphasize the smallness of the effect on the pseudocritical temperature at the beginning of Sec. IV.E and streamlined the discussion on the top left of page 15.

(4) "In Fig 8 it would be helpful to include the crossover line."

Agreed and done.

(5) "In App B it is stated that the deconfinement transition in SU(3) Yang-Mills theory is of second- order in contradiction to other findings."

This was an embarrassing typo which has been corrected.

(6) "Fig 4 the colour coding is hard to read and several word duplications like the the etc appear in the
main body."

In order to highlight how \mu_{S0} depends on both T and \mu_B, we also show many different values of \mu_B. To us, this is more appropriate than a 3D plot. We state in the caption that \mu_B is increasing from bottom to top. Furthermore, a rainbow color scheme is a very common and, in our opinion, easy to read color scheme.
We found some word duplications and removed them.

Referee 2

(1) "Since the authors work in a FRG-improved PQM there are only mesonic and quark degrees of freedom present. This leads me to two points of confusion: (a) What happens at very large temperatures to the pressure and trace anomaly? Does the result for the pressure go to the Stefan-Boltzmann limit for QCD as T -> infty? It would seem to me that the answer is no because there are no dynamical gluons (extra Nc^2 -1 massless degrees of freedom). If I can't go to T-> infty, what is the highest temperature at which I would expect such a model to be trustworthy? (b) What is the impact of throwing out baryons/resonances as one approaches Tc? Traditionally one expects to be sensitive to higher mass states as one approaches Tc."

First of all, we a have a coupled system of quarks, mesons and gluons. Quarks and mesons are dynamical fields while the gluon itself is a background field. However, the effective gluon potential we use as an input also contains, besides the pressure due to gauge field fluctuations, also information on the fluctuations of gluons/Polyakov loops which are encoded in the correlation functions of Polyakov loops (see Ref. [77] in the manuscript). Furthermore, we emphasize that the contribution of quark fluctuations to the Polyakov loop potential is also taken into account, since the the quarks are coupled to the gluon background field. This is the reason why we also see a crossover transition in the Polyakov loops, even though the input potential carries only the first-order transition of the pure gauge theory. What is missing is the dynamic back-reaction of quarks and meson to the gluon sector, which is encoded in the parameters of the Polyakov loop potential U_glue. To effectively take this into account, we use a flavor and chemical potential dependent parameter T_0, which is discussed in App. A. We chose an agnostic approach and avoided labelling this, but it could be called beyond mean-field. However, we believe that our detailed explanation in Sec. II.C and App. A gives a clear account of what is (and is not) done regarding the gluon degrees of freedom.

(a) Since the gluon pressure is encoded in the Polyakov loop potential, the pressure in our work reaches the Stefan-Boltzmann limit at large temperatures by construction. Regarding thermodynamics, the limit T -> \infty is innocuous in PQM models. All this is discussed in more detail, e.g., in the review [34] and corresponding references therein. Since the large temperature limit is not of immediate relevance here, we do not discuss it in the manuscript.

(b) The FRG is based on the fluctuations of off-shell degrees of freedom. Hence, there is no sensitivity to higher mass states. The good agreement with existing lattice results corroborates this statement. We emphasize, however, that while there is a clear hierarchy in the relevance of degrees of freedom in the sense that the lighter the particle the more relevant it is, other quantum numbers, such as charge or strangeness, are certainly relevant for an accurate description, e.g., of net-charge and net-strangeness fluctuations. This aspect is discussed in our subsequent work arXiv:1809.01594. The relevant part of this discussion can be found in the last paragraph of Sec. II.A.

(2) "There are no uncertainties lies in Tables I or II. These were determined by some fitting procedure. It would be nice to know the uncertainties and the impact these uncertainties have on the final results."

This is a very good point. However, the parameters of the Polyakov loop potential are taken from Ref. [77], which do not have uncertainties. We now point out our source in the caption of Tab. II and right after Eq. (A1).

Regarding Tab. I, uncertainties on the parameters shown there would be directly related to the uncertainties in the masses and decay constants we use to fix our free parameters. While we think that it would be worthwhile to explore how the uncertainties in the initial parameters propagate to the final results, our general conclusions will most certainly not be altered and we therefore refrained from such an analysis for the sake of simplicity. To us, this level of quantitative sophistication is only appropriate if the relevant improvements on our approximation discussed at the end of the summary are also taken into account. We mention this now in Sec. IV.A., in the paragraph after Eq. (49).

(3) "In the intro the authors state that they will "go beyond mean field level". This seems to be true for mesonic fluctuations but not for the Polyakov loop itself. This should be stated clearly in the intro elsewhere where it's stated that you are going beyond mean field level."

See our reply to point (1). We also mention that the gluon is a background field in the introduction. We believe that our discussion in Sec. II.C is explicit enough to make clear what is done here regarding the gluons.

In addition to the points raised by the referees, we clarified the definition of the speed of sound in Eq. (54) and Fig. 6. Since we want to compare thermodynamic quantities as functions of T for different \mu_B and \mu_S, our definition yields the speed of sound as a function of T, \mu_B and \mu_S. In hydrodynamics, the speed of sound is defined at fixed s/n_B and n_S/n_B. To avoid confusion, we denote "our" speed of sound as \tilde{c}_S and point out that it agrees with the actual speed of sound only at vanishing density.

---

## Round 2 · List of Changes

(1) Edited Eqs. (13) and (14) to make the dependence on \bar{A} explicit.

(2) Added/modified Eqs. (16) and (17) on page 4.

(3) Removed the wave function renormalization / anomalous dimension from the discussion in and after Eq. (38).

(4) Emphasized the smallness of the effect of strangeness neutrality on the pseudocritical temperature at the beginning of Sec. IV.E and streamlined the discussion on the top left of page 15.

(5) Added the chiral and deconfinement crossover lines in Fig. 8.

(6) Corrected an embarrassing typo in App. B, left column on page 19.

(7) Removed some word duplications that we were able to find.

(8) Added our source to the caption of Tab. II on page 17, and pointed it out again after Eq. (A1)

(9) Mentioned potential errors from the initial conditions in Sec. IV.A., in the paragraph after Eq. (49).

(10) Clarified the definition of the speed of sound in and right after Eq. (54).

---

## Editorial Decision

published